# Above Cloud Aerosol Optical Depth from airborne observations in the South-East Atlantic

Samuel E. LeBlanc[1,2], Jens Redemann[3], Connor Flynn[3], Kristina Pistone[1,2], Meloë Kacenelenbogen[2], Michal Segal-Rosenheimer[1,4], Yohei Shinozuka[5,2], Stephen Dunagan[2], Robert P. Dahlgren[6,2], Kerry Meyer[7], James Podolske[2], Steven G. Howell[8], Steffen Freitag[8], Jennifer Small-Griswold[8], Brent Holben[7], Michael Diamond[9], Robert Wood[9], Paola Formenti[10], Stuart Piketh[11], Gillian Maggs-Kölling[12], Monja Gerber[13], Andreas Namwoonde[14]

[1]Bay Area Environmental Research Institute, Moffett Field, CA
[2]NASA Ames Research Center, Moffett Field, CA
[3]University of Oklahoma, Norman, OK
[4]Department of Geophysics and Planetary Sciences, Porter School of the Environment and Earth Sciences, Tel-Aviv University, Tel-Aviv, Israel
[5]Universities Space Research Association, Columbia, MD
[6]California State University Monterey Bay, Seaside, CA
[7]NASA Goddard Space Flight Center, Greenbelt, MD
[8]University of Hawai`i at Mānoa, Honolulu, HI
[9]University of Washington, Seattle, WA
[10]LISA, UMR CNRS 7583, Université Paris Est Créteil et Université Paris Diderot, Institut Pierre Simon Laplace, Créteil, France
[11]NorthWest University, South Africa
[12]Gobabeb Research and Training Center, Gobabeb, Namibia
[13]Sam Nujoma Marine and Coastal Resources Research Centre (SANUMARC), University of Namibia, Henties Bay, Namibia

*Correspondence to:* Samuel E. LeBlanc (samuel.leblanc@nasa.gov)

## Abstract

The South-East Atlantic (SEA) is host to a climatologically significant biomass burning aerosol layer overlying marine stratocumulus. We present the first results of directly measured Above Cloud Aerosol Optical Depth (ACAOD) from the recent ObseRvations of Aerosols above CLouds and their intEractionS (ORACLES) airborne field campaign during August and September 2016. In our analysis, we use data from the Spectrometers for Sky-Scanning Sun-Tracking Atmospheric Research (4STAR) instrument and found an average ACAOD of 0.32 at 501 nm (range of 0.02 to 1.04), with an average Ångström Exponent (AE) above clouds of 1.71. The AE is much lower at 1.25 for the full column (including below cloud level aerosol, with an average of 0.36 at 501 nm and a range of 0.02 to 0.74), indicating the presence of large aerosol particles, likely marine aerosol, in the lower atmospheric column. ACAOD is observed from 4STAR to be highest near coast at about 12°S, whereas its variability is largest at the southern edge of the average aerosol plume, as indicated by 12 years of MODIS observations. In comparison to MODIS derived ACAOD and long-term fine-mode plume-average AOD along a

diagonal routine track extending out from the coast of Namibia, the directly-measured ACAOD from 4STAR is slightly lower than the ACAOD product from MODIS. The peak ACAOD expected from MODIS AOD retrievals averaged over a long term along the routine diagonal flight track (peak of 0.5) is measured to be closer to coast in 2016 at about 1.5° - 4° E, with 4STAR ACAOD averages showing a peak of 0.42. When considering the full observation set over the SEA, by spatially binning each sampled AOD, we obtain a geographically representative mean ACAOD of 0.37. Vertical profiles of AOD showcase the variability of the altitude of the aerosol plume and its separation from cloud top. We measured larger AOD at high altitude near the coast than farther from coast, while generally observing a larger vertical gap further from the coast. Changes of AOD with altitude are correlated with carbon monoxide, a gas tracer of the biomass burning aerosol plume. Vertical extent of gaps between aerosol and cloud show a wide distribution, with a near zero gap most frequent. The gap distribution with longitude is observed to be largest at about 7°E, farther from coast than expected from previous studies.

# 1 Introduction

Aerosol above clouds have been identified as a leading source of uncertainty in measuring the global source of aerosol burden, constituting globally 25±6% of total burden (Waquet et al., 2013a). In the South-East Atlantic (SEA), where one of the Earth's semi-permanent stratocumulus cloud decks exists, the frequency of occurrence of an overlying aerosol layer averaged over the entire region is more than 30% on an annual basis, and increase to more than 50% during the peak biomass burning season of July through November (Devasthale and Thomas, 2011, Zhang et al., 2016). These aerosols above clouds impact climate by either directly affecting the radiative budget (e.g., Schulz et al., 2006), by interacting with clouds via a change in the atmospheric thermal profile (semi-direct effects) (Sakaeda et al., 2011), or by directly modifying cloud properties (indirect/Twomey effect) (Bond et al., 2013; Twomey, 1974). One of the driving uncertainties in quantifying the impact of these aerosols is due to the difficulty in retrieving the Above Cloud Aerosol Optical Depth (ACAOD) from satellite measurements, where the ACAOD is the Optical Depth of the aerosol layers that are present at higher altitudes than the cloud tops. To constrain the climatic effect of the aerosol above cloud in the SEA, an airborne field campaign, ObseRvations of Aerosols above CLouds and their intEractionS (ORACLES), was conducted in the peak of the biomass burning season (ORACLES Science Team, 2017) in conjunction with other large scale field missions focused in the same region; CLARIFY (CLoud – Aerosol – Radiation InteRactions and Forcing for Year 2017; Zuidema et al., 2016), AEROCLO-sA (AErosols, RadiatiOn and CLOuds in southern Africa; Formenti et al., 2019), and LASIC (Layered Atlantic Smoke Interactions with Clouds; Zuidema et al., 2018). We show in this paper the directly measured ACAOD and its vertical dependence during the first phase of ORACLES.

Although much progress to quantify aerosols above clouds has been made, direct measurements of the ACAOD in the SEA is limited. Previous measurements during the Southern African Regional Science Initiative Project (SAFARI-2000) sampled only small, near coast portions of the overlying aerosol layer with limited instrumentation (Keil and Haywood,

2003; Bergstrom et al., 2003). To date, several passive satellite sensors (e.g., Moderate Resolution Imaging Spectroradiometer (MODIS), Polarization and Directionality of Earth's Reflectances (POLDER), Ozone Monitoring Instrument (OMI)) have been used to detect aerosol above clouds and retrieve ACAOD over the SEA region (e.g., Jethva et al (2013, 2014), Waquet
et al. (2009, 2013b), Torres et al. (2012), De Graaf et al. (2012, 2014), Meyer et al. (2015), Peers et al. (2015), Feng and Christopher (2015), Sayer et al. (2016), Chang and Christopher (2016, 2017)). However, current passive satellite ACAOD retrieval techniques could be biased by what is called the "cloud adjacency effect" (Wen et al., 2007) or the "3-D cloud radiative effect", i.e., brightening of cloud-free air near clouds, that also extends to above cloud aerosol
properties, which has been observed using polarized light (Cornet et al., 2018). 3-D cloud radiative effects also impact retrievals of aerosol above clouds, where the underlying cloud heterogeneity impact the aerosol subjected radiance (Peers et al., 2015).This is why some studies have used active sensors such as CALIOP (Cloud Aerosol LIdar with Orthogonal Polarization) instead of passive satellite sensors to retrieve ACAOD (e.g. Hu et al., 2007; Chand
et al., 2009; Wilcox, 2012; Matus et al., 2015; Zhang et al., 2014; Kacenelenbogen et al., 2019). We refer the reader to Table 1 of Kacenelenbogen et al. (2019) for a more complete list of passive and active satellite sensors used in the observation of ACAOD over other parts of the world.

Underlying assumptions of aerosol optical and microphysical properties and of cloud properties in retrievals of ACAOD from satellites can lead to large uncertainties or biases. Examples of assumptions include: a constant spectral aerosol absorption, which has the largest influence on retrieval uncertainty (e.g., Chand et al., 2009, Meyer et al., 2015); aerosol properties don't vary over a large spatial region, or are representative of all aerosol over large regions, and have a
constant vertical dependence (e.g., Torres et al., 2012); retrieved aerosol properties over highly reflective and opaque clouds are representative of all aerosol (e.g., Hu et al., 2007, Peers et al., 2015); and/or the impact of aerosol absorption on polarized reflectances can be neglected (e.g., Waquet et al., 2013b, Peers et al., 2015). Active remote sensors also have issues in retrieving ACAOD, due to low signal to noise ratio of aerosol backscatter attenuated by overlying
aerosols, as demonstrated for CALIOP during daytime (e.g., Hu et al., 2007, Deaconu et al., 2017). The ACAOD presented here does not suffer from these common retrieval assumptions as it is directly measured with an airborne sunphotometer and can be used to calibrate/validate satellite retrievals of ACAOD (e.g., Sayer et al., 2019).

Not only is the climatological magnitude of the ACAOD in question, but its vertical dependence and relative distribution with respect to clouds are uncertain as well. Distinct Clear-Air-Slots (CAS) separating aerosol and cloud layers were first reported by Hobbs (2003). A separation of the cloud and aerosol layers indicates that aerosols are not directly modifying cloud microphysical properties (e.g., Twomey 1977), but rather directly modify the radiation field and
semi-directly the underlying clouds (e.g., Graßl, 1979, Lohmann and Feichter, 2005), or that clouds previously processed and depleted overlying aerosols. Past work has shown that the elevated aerosol layers in this region are frequently separated from the underlying cloud top. Devasthale and Thomas (2011) found that 90-95% of above-cloud-aerosol cases observed by CALIOP (which has known limitations, e.g. Kacenelenbogen et al., 2014) showed a gap larger

than 100 m. Rajapakshe et al (2017) showed ~40% incidence of a gap between cloud top and aerosol layer bottom as measured by the spaceborne lidar Cloud-Aerosol Transport System (CATS; McGill et al., 2015), of which 60% have a gap of less than 360 m. Additionally, the gap is expected to be dependent on the distance from coast, decreasing further from coast, with a few examples of situations without a gap between cloud and aerosol, as observed by CALIOP (Sakaeda et al., 2011; Wood et al., In prep; Deaconu et al., 2019). The differences between these estimates on the presence of the CAS, can be refined through direct airborne sampling, as during ORACLES.

In Section 2, we present an overview of the first ORACLES deployment and introduce the instruments and related data quality. Section 3 details some of the methodology for specific analysis. Section 4 presents the measurements of ORACLES ACAOD, their spatial and spectral dependence, and a comparison to ACAOD climatologies derived from MODIS satellite measurements. Additionally, in Section 4 we show some advanced analysis from the airborne sunphotometer with the vertical dependence of ACAOD and the measured gap between the aerosol layer and the clouds. The summary of our results is presented in Section 5. An appendix describes the 4STAR instrument's data processing methodology and data quality.

## 2 Data and instrumentation

We focus on the Aerosol Optical Depth (AOD) measurements from the Spectrometers for Sky-Scanning Sun-Tracking Atmospheric Research (4STAR; Dunagan et al., 2013) airborne sunphotometer on board the NASA P-3 during ORACLES 2016. For additional context, we use a combination of in situ instrumentation providing aerosol optical properties, cloud particles, and trace gas measurements. We also use nearby AERONET (Aerosol Robotic Network; Holben et al., 1998, 2018) stations, and regional satellite AOD data for spatial context and comparisons. Satellite measurements give context by either a long-term record using neighboring clear sky AOD retrieval from the Moderate resolution Imaging Spectroradiometer (MODIS; Levy et al., 2013) or a short-term record using the newly developed retrieval of ACAOD from MODIS (Meyer et al., 2015).

### 2.1 ObseRvations of Aerosols above CLouds and their intEractionS (ORACLES)
The ORACLES field campaign is aimed at directly measuring the SEA ACAOD and its direct, indirect, and semi-direct radiative effects on climate via airborne sampling during 3 intensive operating periods (Sep 2016, Aug 2017, Oct 2018) (Zuidema et al., 2016). The NASA P-3 flew as an airborne platform for in situ and remote sensing measurements of aerosols and clouds in all three campaigns, along with NASA ER-2 high altitude remote sensing platform in 2016 only. The 2016 deployment out of Walvis Bay, Namibia, included 15 successful flights for the P-3 from August 27 to September 29 (ORACLES Science Team, 2017). Nearly half of these research flights followed a routine flight path extending diagonally from 13°E, 23°S to 0°E, 10°S and the other half focused on paths with increased chance of successful sampling with all instruments (see Fig. 1). All flights (P-3 and ER-2) were planned using the research flight planning software developed by LeBlanc (2018).

## 2.2 Spectrometers for Sky-Scanning Sun-Tracking Atmospheric Research (4STAR)

The 4STAR instrument determines in-flight aerosol optical depth (AOD) from airborne measurements of direct solar radiation. 4STAR incorporates a modular sun-tracking/sky-scanning optical head protruding above the aircraft fuselage, an instrument rack within the aircraft cabin housing a computer, motion control, and two grating spectrometers, and an electrical umbilical and fiber optic cable connecting the optical head and the rack. This airborne sun tracker and sky radiometer has multiple operating modes (direct sun, sky scans (Pistone et al., 2019), and zenith under cloud (LeBlanc et al., 2015)), which are selected by an operator depending on the sky conditions. Using 2 spectrometers, 4STAR records hyperspectral radiation measurements spanning the continuous wavelength range from 350 nm to 1750 nm, with spectral resolution of 2 - 3 nm below 1000 nm and 3 - 7 nm at longer wavelengths. These hyperspectral radiation measurements yield AOD over the continuous wavelength range, broken only by prominent gas absorption lines. The full width of the field of view for the direct beam irradiance measurement is 2.4° with radiometric deviations of less than 1% across this span. The nominal calibration accuracy of AOD measurements from 4STAR are dependent on wavelength, time of day, tracking stability, stability of radiometric calibration, and various second-order corrections (such as removal of light absorption by trace gases). The accuracy is typically near 1% in transmittance (at 500 nm) resulting in an AOD uncertainty of 0.01 at solar noon. The details on the calibration, corrections and uncertainty assessment of 4STAR AODs are found in the appendix.

## 2.3 In situ instrumentation: HiGEAR, PDI, and COMA

A combination of in situ instruments is used to provide context for the AOD measurements.  We use aerosol scattering from nephelometers from the Hawaii Group for Environmental Aerosol Research (HiGEAR), cloud droplet number concentration from the Artium Flight Probe Dual Range Phase Doppler Interferometer (PDI), and CO concentration from CO Measurements and Analysis (COMA), as described below.

We use the aerosol scattering coefficient at 550 nm, corrected for ambient outside relative humidity, which is calculated from nephelometers measurements operated as part of the HiGEAR extensive airborne measurement suite (similar to Howell et al. 2006). These nephelometers directly ingest aerosol from ambient air, and together with other HiGEAR instrumentation provide size resolved assessment of aerosol physical and chemical properties and their relationship to measured optical and microphysical behavior. The scattering coefficient of the aerosol is sampled with 3-wavelength nephelometers (TSI 3563, at 450 nm, 550 nm, and 700 nm) while dependence on humidity is measured with paired single-wavelength nephelometers (two - Radiance Research M903 measuring at 540 nm; one  with air humidified to 80% relative humidity, and  the other did not control the RH). Comparisons between the dry Radiance Research to the TSI nephelometers are used to correct the Radiance Research truncation issues, while the humidity dependence of the scattering coefficient is calculated from a gamma relationship obtained from the paired Radiance Research nephelometers (following Quinn et al., 2005). We also use an extinction coefficient at 550 nm, which is calculated from the corrected scattering and measured absorption coefficient. The absorption coefficient is

measured in dry conditions using Particle Soot Absorption Photometers (PSAP) from Radiance Research. The solid diffuser inlet efficiently samples particles <1μm, with a 50% cutoff at approximately 3 μm (McNaughton et al., 2007).

Cloud drop concentration was sampled from the PDI, mounted on a wing pylon of the NASA P-3. The PDI uses interferometry with a diagnostic technique for sampling cloud droplet size and velocity at the same time (e.g., Chuang et al., 2008, Small et al., 2009). The combined range of 2 lasers with differing wavelengths covers liquid cloud droplets sized 1 to 1000 μm or larger.

CO concentration from the in situ sampled air is reported using the COMA instrument, which includes the ABB/Los Gatos Research $CO/CO_2/H_2O$ Analyzer modified for flight operations. It uses off-Axis ICOS technology to make stable cavity enhanced absorption measurements of CO, $CO_2$, and $H_2O$ in the infrared spectral region, technology that previously flew on other airborne research platforms with a precision of 0.5 ppbv over 10 seconds (Provencal, et al.,
2005; Liu, et al., 2017).

## 2.4 Local AERONET stations
New AERONET stations were set up to give context to ORACLES measurements in south-western Africa along with two pre-existing stations in Namibia, neighboring the SEA. In addition
to the new permanent sites, the highly spatially resolved DRAGON (Distributed Regional Aerosol Gridded Observation Networks (Holben et al., 2018)) network of 6 AERONET stations were located near Henties Bay, about 100 km north of the NASA P-3 base station of Walvis Bay, Namibia, for the duration of ORACLES 2016. In addition to these stations, we use the data from the stations located at Walvis Bay Airport, Gobabeb, and Henties Bay in Namibia, and
Lubango and Namibe in Angola. The reported data from these AERONET sites and DRAGON represent the entire span of available sampled full column AOD during the deployment time range, including potential local sources. To focus on the smaller aerosol of the lofted biomass burning aerosol (e.g., Pósfai et al., 2003) and reduce the influence of local sources such as large dust and sea salt aerosol particles, we report the fine mode AOD, derived using the
Spectral Deconvolution Algorithm (O'Neill et al., 2003).

## 2.5 Satellites and climatology
Recent advances in satellite imager retrieval methodology enables the use of MODIS spectral cloud reflectances to obtain the overlying aerosol optical properties jointly with the cloud optical
properties (Jethva et al., 2014; Meyer et al. 2015; Sayer et al., 2016). The algorithm used here, MOD06ACAERO (Meyer et al., 2015), simultaneously retrieves above-cloud AOD and the cloud optical thickness and effective radius of the underlying marine boundary layer clouds while also providing pixel-level estimates of retrieval uncertainty that accounts for known and quantifiable error sources (e.g., radiometry, atmospheric profiles, and cloud and aerosol radiative models).
MOD06ACAERO uses reflectance observations at six MODIS spectral channels from the visible to the shortwave infrared. Retrievals are run on both Terra (morning) and Aqua (afternoon) MODIS instruments with a constant aerosol-cloud vertical geometry and two different aerosol intrinsic property model assumptions. The aerosol models stem from either Haywood et al. (2003) or from the standard MODIS Dark Target land Aerosol product, which is the model used

in this work (MOD04; Levy et al., 2009). The cloud forward model, ancillary data, and other retrieval assumptions are consistent with those of the operational MODIS cloud products (MOD06) (Platnick et al., 2017). Meyer et al. (2015) showed MOD06ACAERO retrieved cloud optical thicknesses and effective radius are consistent in range and values with the standard MODIS cloud products, and larger than the standard above cloud AOD product from the spaceborne CALIOP. Consistent with Meyer et al. (2015), we report only the AOD from MOD06ACAERO above clouds with an optical thickness of greater than 4, and AOD uncertainties lower than 100%. Note also that for this work the retrievals are aggregated to a 0.1° equal-angle latitude/longitude grid.

For another comparison, we use the standard Dark Target aerosol retrieval from MODIS clear sky pixels in the SEA that has been retrieving aerosol properties from reflectances measured since 2001 (Levy et al. 2013). We used 12 years of the high-resolution time series of the MODIS retrieved fine mode AOD sampled during August and September as a proxy for an ACAOD climatology similarly to Zuidema et al. (2016). Using the fine-mode total column AOD to represent the smoke aerosol above cloud in this region is supported by the aerosol's typically small size (Pósfai et al., 2003), and is used to exclude the coarse mode aerosol which mostly consists of boundary layer sea salt and dust along the coast. The presence of biomass burning aerosol results in the fine-mode fraction vastly dominating the optical characteristics of above cloud aerosol in the region (e.g., Yoon et al., 2012, and fine mode fraction by volume in Russell et al., 2014). When there is a significant amount of biomass burning aerosol in the boundary layer in addition to the aerosol above cloud, this fine mode assumption is expected to be an overestimate.

### 3 Methodology

### 3.1 AOD above cloud determination

During ORACLES, we sampled multiple types of scenes, some of which were described by Hobbs (2003), which had CAS (i.e., described in this paper as gaps) within aerosol layers and between aerosol and cloud layers. Some scenes show a gap between the aerosol layer and the clouds, some show no gap, and some show a gap between two aerosol layers. Examples of these cases have been collected via photography from the NASA P-3 and are shown in Fig. 2, similarly portrayed by Hobbs (2003). These photographs were selected for easier visual identification, although not always showing scenes with 100% cloud cover. Aerosol appears visually darker than the background light blue sky when the observer is at or below the altitude of the aerosol layer (Fig. 2a & 2b). When the aerosol appears directly above clouds, it can be interpreted as a lighter colored haze extending from cloud top, sometimes making it harder to distinguish between aerosol and cloud boundaries (Fig. 2c).

The AOD measurements that are used to quantify the aerosol above cloud in the presence of a gap, can extend thousands of meters vertically, because the aerosol within a gap contributes minimally to the overall ACAOD. For conditions without gaps, where the lowest aerosol layer is touching the top of the cloud, the ACAOD is measured when the aircraft is immediately above cloud. To identify the measurements where 4STAR sampled ACAOD (including AOD

measurements within a gap), we start with the periods of flights defined by the P-3 module flags as legs directly above cloud. These P-3 module flags were created using manual inspection of flight altitude time series and flight scientist mission notes from every flight (Diamond et al., 2018). We supplement these module flags with a manual inspection of the AOD time series

from 4STAR, and select each sample measured directly above a cloud layer and up to the bottom of the aerosol layer. The cloud layer was defined by a cloud drop concentration greater than 10 cm$^{-3}$ as measured by the PDI. When the PDI was not operational, we used lack of sun tracking from 4STAR, high outside ambient relative humidity (>90%), and/or visual inspection of in-flight video as the metric for being in clouds. The bottom edge of the aerosol layer is defined

at the altitude that has a 10% change in AOD and a dry scattering coefficient at 550 nm of either 50 Mm$^{-1}$ or a change by more than 75%. Figure 3 shows profiles with color-coded vertical regions to demonstrate the selection of the ACAOD portion of the AOD measurements.

### 3.2 Ångström Exponent (AE) calculations

The relationship of the AOD at various wavelengths is used to determine the Ångström exponent (AE, or sometimes referred to as the *extinction* Ångström exponent) (Ångström, 1929), which is inversely related to the size of the aerosol particles. The AE for the sampled AOD is not only dependent on the size distribution of aerosol particles but also on the type of aerosol measured (e.g., Russell et al., 2014). As a first approximation, large aerosol particles

will typically have small AE values and small aerosol particles will have large AE values (e.g. an AE value between 0.1 and 1 for large marine aerosols (Sayer et al., 2012) or above 1.5 for small biomass burning or urban industrial aerosols (Russell et al., 2014, Fig 6, LeBlanc et al., 2012)). According to Dubovik et al., (2002), AERONET-derived AE values (computed between 440 and 870nm) for biomass burning aerosols are between 1.2 and 2.1 in Bolivia or Brazil, whereas AE

values from desert dust aerosol are between 0.1 and 0.9 in Saudi Arabia. The AE measured in the source regions of the biomass burning from SAFARI-2000 showed a range between 1.6 and 2.1 from Mongu, Zambia during the biomass burning season (Eck et al., 2003). Here we evaluate AE using two methods: 1) by fitting a second-order polynomial to the logarithm of the AOD spectra from selected wavelengths between 355 nm to 1650 nm and finding its derivative

at any one wavelength, (here at 500 nm, AE$_{500}$) (similar method to O'Neill et al., 2001; Shinozuka et al., 2011), and 2) the negative of the slope of the AOD with wavelength in logarithmic scale (two wavelengths used here 470 nm and 865 nm, AE$_{470/865}$) (e.g. Dubovik et al., 2002).

When AOD spectra are not a straight line in a log-log plot but rather slightly curved, this

indicates that the AE is wavelength dependent. The curvature of AE (spectral dependence of the AE) is related to the aerosol size distribution (e.g., Kaufman, 1993, Eck et al., 1999, O'Neill et al., 2001, Yoon et al., 2012) and additionally to the aerosol absorption (Kaskaoutis and Kambezidis, 2008). The two methods to calculate AE (evaluated at different wavelengths) can be used to quantify the AE curvature and refine the aerosol size distribution or fine mode

fraction (e.g., Yoon et al., 2012).

# 4 Results and discussion

4.1 Statistics of sampled ACAOD and spatial distribution

We have separated all 4STAR measurements in the SEA into either ACAOD (11.5 hours of measurements, from flags described in section 3.1) or full column AOD (0.9 hours of measurements in level legs or profiles below 600 m in altitude). The full column AOD is distinct from the ACAOD measurements as they necessarily require conditions without overlying cloud and thus will include the elevated biomass burning layer as well as any lower-level aerosol near the sea surface. We note that these two populations do not necessarily coincide directly in space and time, but may be combined in a statistical sense. Figure 4 shows the distribution of those measurements, with roughly 1 sample per second, at two wavelengths. The ACAOD at 501 nm (ACAOD$_{501}$) from all samples (blue bars) has a mean, median, and standard deviation of 0.32, 0.33, and 0.15 respectively, with an absolute range of 0.02 to 1.04. The full column AOD (pink bars) has a mean, median, and standard deviation of 0.36, 0.30, and 0.18, respectively, with an absolute range of 0.02 to 0.74. The larger mean AOD are likely representative of the combined aerosol burden from within the boundary layer as well as the typical plume observed aloft, although exhibiting larger variability as shown by the larger standard deviation. The smaller range of values for the full column AOD as compared to ACAOD, is likely caused by the lower number of full column AOD measurements and their differences in location and time compared to the ACAOD measurements. The small difference between the mean above cloud and full column AODs indicates that the majority of the AOD$_{501}$ sampled in the region is due to the elevated layers of aerosol. In contrast, the AOD sampled at 1020 nm (AOD$_{1020}$) is much larger for the full column than its above cloud counterpart by nearly 70%, with the full-column AOD$_{1020}$ having a mean, median, and standard deviation of 0.15, 0.13, and 0.06 respectively, and the ACAOD$_{1020}$ at 0.09, 0.09, and 0.05 respectively with a range of 0.01 to 0.75 (Fig. 4b).

Considered together, the ACAOD and full column AOD (denoted by the total extent of the histogram bars in Fig. 4) represent what a satellite remote sensor would retrieve in the region, if it were spatially and temporally co-located to the NASA P-3 aircraft and if the retrievals would not discriminate between full column and over clouds. The mean, median, and standard deviation of AOD$_{501}$ for all combined measurements is 0.32, 0.33, and 0.15, respectively, though we note that this is dominated by the greater sampling of ACAOD (N=39229) vs full column AOD (N=3395). The uncertainty in ACAOD sampled by 4STAR due to instrumental artifacts and calibration (see appendix for more details) is 0.011, 0.01, and 0.008 (0.013, 0.012, and 0.012) for the average, median, and standard deviation, respectively, at 501 nm (1020 nm).

The spatial distribution of the ACAOD$_{501}$ is presented in Fig. 5. The ACAOD was averaged in nearly equidistant latitude and longitude bins (0.65° latitude by 0.6° longitude). We observe highest ACAOD near the western coast of Africa at the northernmost parts of the sampled region, while the lowest ACAOD is in the south of the sampled region. The higher ACAOD extends to the west but at reduced AOD compared to near coast, consistent with the expected behavior of the climatological plume (Fig. 1 and Zuidema et al., 2016). The higher average ACAOD in the northernmost part of the sampled region is also observed in the fine mode AOD

from ground based AERONET stations along the southern African coast (triangle symbols in Fig. 5).

The variability in standard deviation shows that, in the north, variability is low in measured
ACAOD (Fig. 5b). Note that the standard deviation here is calculated as a fraction of all samples, and we show the total number of flight days contributing to each bin to give context as to the temporal variability observed. The largest variability of the sampled ACAOD seems to be concentrated in the central portion of the measured region, around 18°S and 8°E, with ACAOD standard deviation exceeding 0.15, over the 3 - 5 days sampled. This high variability is
consistent with a day-to-day change in the location of the southern edge of the highest AOD in the aerosol plume climatology for September (Fig. 1 and Zuidema et al. 2016). Large variability is also observed near Walvis Bay, Namibia, outside the typical climatology for the biomass burning plume. This variability in ACAOD is likely caused by local production of aerosol, observed to be mostly dust or large particles. This hypothesis is corroborated with ground-
based measurements from an AERONET station located at the Walvis Bay Airport which shows a large but variable coarse mode fraction of AOD (average 58%±19% of coarse mode fraction), and consistently larger aerosol effective radius from sky scan retrievals. The fine mode fraction of the AOD sampled by AERONET near the Walvis Bay Airport also shows some variability (Fig. 5b), but this is dwarfed by the coarse mode variability (not shown).
The full column $AOD_{501}$ sampled by 4STAR and AERONET locations is presented in Fig. 5c, where its paucity of samples is apparent particularly in the central sampling region where ACAOD shows higher than average values. The occasions where the P3 sampled the full column AOD occurred nearly always at the edges of the cloud layers. These full column
measurements were not inside pockets of open cells clouds (POC; Stevens et al., 2005; Wood et al., 2011). Full column AOD measurements were more commonly measured past the southern edge of the stratocumulus cloud deck, and where the marine boundary layer was both polluted by biomass burning or with a clean background (ORACLES Science Team, 2017). Where a direct comparison of the full column AOD and the ACAOD is possible, the full column
$AOD_{501}$ is higher by an average of 0.03 (mean full column $AOD_{501}$ is 0.38 vs. mean $ACAOD_{501}$ is 0.35 at the same locations). This difference is nearly reproduced by AERONET, impacted by dust and sea salt in the boundary layer over land with overlying biomass burning aerosol, in the average fine mode $AOD_{501}$ (0.2) and total $AOD_{501}$ (0.24)

An average ACAOD of this region can be calculated from these binned spatial statistics, representing a more even weighting of the ACAOD (equal spatial bins) as compared to averaging over the total number of samples which could be influenced by variability in sampling density. This averaging method attempts to reduce the spatial sampling bias from sampling the same area multiple times (like for the relatively low ACAOD near Walvis Bay), but at a cost of
temporal resolution. The mean $ACAOD_{501}$ and its mean uncertainty is 0.37±0.01, which is arguably more representative of the SEA region, as determined by the average of the mean within each spatial bin. The median $ACAOD_{501}$ and median uncertainty of the region is 0.34±0.01 and the average standard deviation and the uncertainty's average standard deviation

is 0.05±0.004. The equivalent spatially averaged, median, and average standard deviation of $ACAOD_{1020}$ is 0.11±0.02, 0.09±0.01, and 0.02±0.004.

## 4.2 Spectral AOD above cloud and its Ångström Exponent

The spectral characteristics of ACAOD is related to the aerosol intensive properties (shape, size distribution, absorption, and refractive index) (e.g., Kaskaoutis and Kambezidis, 2008, O'Neill et al., 2001). From all measurements of ACAOD at wavelengths outside strong gas absorption, we created ACAOD spectra representing the mean, median, and related standard deviation (Fig.

6), which is representative of the sampled ACAOD throughout this deployment. This ACAOD spectra is consistent with the mean, median, and standard deviation of the ACAOD at 501 nm and 1020 nm presented in Fig. 4. The ACAOD spectra for both the mean (0.38 at 452 nm; 0.13 at 865 nm) and median (0.38 at 452 nm; 0.12 at 865 nm) are easily within the mean uncertainty (0.013 at 452 nm; 0.008 at 865 nm) of the measured spectra. The standard deviation of the

ACAOD (0.18 at 452 nm; 0.06 at 865 nm) is nearly equivalent to its mean at the longest wavelengths (longer than 1600 nm). This larger standard deviation at longer wavelengths can be caused by sporadic larger AODs at those longer wavelengths, agreeing with the notion of intermittent presence of dust or marine aerosol, or alternatively, this may be linked to lower signal to noise ratio of the 4STAR spectrometers. From the AE information, we can have a

sense of the particle size, but we can have educated insight of aerosol type with the accompanying measurements and prior information for the region. To separate aerosol type (dust or sea salt), a more advanced aerosol classification method would be needed, such as the pre-specified clustering method described by Russell et al. (2014), which used wavelength dependent Single Scattering Albedo and Refractive Index.

There is a distinction between mean AE from ACAOD vs. from full column AOD observed for both methods, $AE_{500}$ and $AE_{470/865}$, described in Sect. 3.2. The mean $AE_{500}$ for ACAOD and full column are 1.45 and 1.08, while the mean $AE_{470/865}$ are 1.71 and 1.25, respectively (see blue and pink solid lines in Fig. 7). The distribution of AE in Fig. 7 seems to indicate that most of the

ACAOD is influenced by fine-mode aerosol particles, which is consistent with aerosol that are aged biomass burning as reported by Eck et al. (1999) and with the aerosol in situ sizing measurements taken on board the NASA P3 (albeit there are inlet passing inefficiencies for accurately sampling larger aerosol (Pistone et al., 2019)). Even though the differences between full column AOD and ACAOD at 501 nm is small, the higher relative difference at 1020 nm

significantly modulates the AE for above cloud and full column. This is consistent with the notion that even a relatively small population of larger aerosol particles (in this case likely sea-salt), has a large impact to the AE, because of their larger AOD in the longer wavelengths (e.g., Yoon et al. 2012).

The difference in average AE evaluated at different wavelengths, $(AE_{500} - AE_{470/865})$ is -0.26 for the ACAOD, which is very similar to the combination of $AE_{470/865}$ and AE difference (centered at an AE difference of -0.2, and $AE_{470/865}$ of 1.85) sampled by the Mongu AERONET station within the biomass burning source region of southern Africa (Yoon et al., 2012). The full column average AE difference of -0.17 with an $AE_{470/865}$ of 1.25 is typical of coarse-mode dominant, with

Mie theory predicting 30% – 40% of fine mode fraction for this combination of AE difference and AE values (Yoon et al., 2012). This large coarse-mode fraction is corroborated by the in situ measurements of large marine aerosol particles during the boundary layer flight segments during ORACLES, or reports of local dust in the boundary layer sampled at the AERONET Mongu station.

The spatial patterns (Fig. 8) of the above cloud $AE_{470/865}$, calculated from each AOD measurement, help indicate the potential changes in aerosol intensive properties measured during ORACLES 2016. For the sampled region, the spatial mean $AE_{470/865}$ ($AE_{500}$), obtained by averaging the mean of each bin over the entire region, is 1.65 (1.44), with a spatial average of the medians is 1.66 (1.48), and a spatial average of the standard deviation within each bin of 0.10 (0.06). This same spatial averaging method was also used in Section 4.1. The spatial statistics of $AE_{470/865}$ and $AE_{500}$ for the full column AOD is lower than its ACAOD counterpart by 0.4 for the mean and by 0.3 for the median, with similar standard deviations. The smallest $AE_{470/865}$ (similarly for $AE_{500}$, not shown) is observed in locations near coast in the southern part of the sampling region and south of the routine flight paths. A distinctively smaller than average $AE_{470/865}$ value is also observed near Walvis Bay, Namibia. This low $AE_{470/865}$ may be coincident with dust or marine aerosol within the sampled column of ACAOD at altitudes of 300 to 3700 m. Further from the coast, there is a small tendency towards decreasing AE values, present in multiple flights, from about 1.8 to 1.6 at 5°E to 3°E, as compared to similar latitudes near-coast. At those same locations (not shown), the $AE_{500}$ of the above cloud aerosol does not show a similar trend, possibly indicating a change in aerosol composition and size. There is however a trend of higher $AE_{500}$ near the center of the region (7°E to 11°E and 20°S to 15°S), by more than 0.2 as compared to the furthest west points. Similar to the map of the standard deviation of the ACAOD (Fig. 5), a larger standard deviation in AE is observed near 18°S and 8°E (Fig. 8), at the variable southern edge of the climatological mean aerosol plume in an area with multiple sampling days. The high standard deviation in AE in this region is associated with ACAOD between 0.2 and 0.45 with AE from 0.2 to 1.2. These aerosols, sampled over more than one day, may not be uniquely biomass burning, but the low AE may indicate that there is water vapor condensation on aerosol by neighboring mid-level clouds, observed in a few flights in that region. Further northwest, a nearly equivalent number of days were sampled, but the standard deviation of the $AE_{470/865}$ is lower, indicating lower day-to-day variability. In the northern near coast region, there are multiple bins that were sampled during only one day; here the standard deviation should not be taken to represent the actual variability of the aerosol, but rather of the sampling accuracy within a day.

## 4.3 Airborne AOD in context of climatology and satellite measurements

To contextualize the ACAOD sampled during the ORACLES 2016 measurements, we compared the ACAOD measured directly below the aerosol layers from the NASA P-3 to those retrieved from MODIS satellite measurements (both standard aerosol Dark Target and above cloud retrievals). We focus on the diagonal routine flight paths (southeast to northwest), where the P-3 sampled the same locations numerous times over the course of the month-long deployment, and the MODIS pixels within 15km of the P-3 tracks. The sampled ACAOD for

each of the routine flights (identified by their day in Fig. 9a) is compared to its equivalent above cloud aerosol retrieved from the combination of MODIS sensors from Aqua and Terra using the MOD06ACAERO methodology described by Meyer et al. (2015) (Sect. 2.5). When comparing ACAOD from 4STAR and MOD06ACAERO for each sampling day, a general agreement for most days is observed with some high deviations at certain longitudes for MOD06ACAERO, albeit with day-to-day variability as to the direction of the agreement. For example, MOD06ACAERO was high compared to 4STAR measurements on 12 September near 7°E, and higher than average ACAOD was measured by both 4STAR and MOD06ACAERO near 3°E on 31 August and 4 September.

We compile daily 4STAR ACAOD and MOD06ACAERO values to a mean and median (spanning the August - September 2016 ORACLES deployment period), which we then compare to a proxy of ACAOD climatology based on the standard MODIS Dark Target fine mode aerosol retrieval (Fig. 9b & 9c). The ACAOD proxy is the monthly-averaged MODIS fine mode AOD for clear-sky pixels that have been aggregated from its original high resolution to 1° in latitude and longitude following the diagonal routine flight track of the P-3. The above cloud aerosol is fine-mode dominant (Sect. 4.2), while the boundary layer aerosol is coarse mode dominant. The general longitudinal dependence and magnitudes of the mean ACAOD as measured by 4STAR are consistent with the MODIS fine mode climatology, with larger ACAOD in the western region (Fig. 9b).

The peak in this climatology occurs near 1°E along the diagonal, whereas 4STAR ACAOD broadly peaks closer to 3°E, and MOD06ACAERO subsampled to routine flights is closer to 2°E. The larger mean MOD06ACAERO at 7°E as compared with 4STAR and the climatology is likely due to anomalously high days skewing the mean (such as 12 September). On the eastern end, between 10° - 12°E, 4STAR measured much lower ACAOD (below 0.1) than the climatology and MOD06ACAERO, but measured higher ACAOD (0.27) at the easternmost edge of the routine flight path, near 14°E. The easternmost 4STAR measurements are within 0.05 of the averages from AERONET ground based measurements over the same routine flight days, which is higher by ~0.15 than monthly averages from AERONET measured during August or September 2016. For the entire longitude span investigated here, 4STAR ACAOD averaged 12.2% lower than the climatology (difference of 0.04 AOD), and 16.0% lower than MOD06ACAERO for September (12.1% of the August mean) along the routine flight track.

The longitudes with the smallest difference between the subsampled MOD06ACAERO and the monthly averages shows where the flight sampling is adequate to represent monthly mean, whereas for regions with large differences, the sampled ACAOD is not representative of its monthly-mean. The peak mean ACAOD for all August and September MOD06ACAERO at the most western edge of the region, near 0°E, is shifted to the east in the mean MOD06ACAERO subsampled for routine flights.

The largest differences between the monthly mean MOD06ACAERO for September 2016 and the subsampled MOD06ACAERO (around 2°E, 6°-7°E, and 10°E), suggest that sampling in that region is not representative of the monthly mean. There is good agreement between the

MOD06ACAERO subsampled and the monthly mean in other longitudes (within 0.05) suggesting that 4STAR ACAOD can be compared to monthly statistics at those locations. In these locations, 4STAR ACAOD had a bias of about 0.05 – 0.08 for most of the flight tracks (4STAR being lower than the subsampled and monthly mean MOD06ACAERO). There is a divergence near coast (12°E) between 4STAR ACAOD and MOD06ACAERO, showing a longitudinal trend in this bias by greater than 0.1.

Similar longitudinal dependence of ACAOD is observed in the medians as with the means, but with greater differences at most longitudes between 4STAR ACAOD and MOD06ACAERO. Differences between the mean and the medians are shown here to reduce impact of outliers in our sparsely sampled data. The MODIS fine mode climatology medians peaks twice in the western edge, near 1°E and 4°E, whereas the measured 4STAR ACAOD peaks at 1°E, and MOD06ACAERO also peaks at 1°E, and again at 7°E, like its means. Median and mean differences for both MOD06ACAERO and 4STAR seem to move their respective maximum further west, and increase matching further east (notably at 9°E), indicating a changing ACAOD distribution with longitude.

Overall, the ACAOD sampled by 4STAR is slightly lower than the MOD06ACAERO counterpart for averages and medians over the same days, additionally, it is lower than the MODIS AOD fine mode climatology. The peak for September 2016 was more eastward than what the MODIS AOD fine mode climatology indicates, with 4STAR measurements peaking even more east than MOD06ACAERO. This shift in peak ACAOD is likely related to differences in meteorology and associated wind patterns or a shifting of the biomass burning source locations for September 2016 as compared to the 12-year climatology. The assumption that all fine mode AOD in clear sky retrieved by MODIS over 12 years is representative of the above cloud AOD should be revisited, as this assumes that 1) no aerosol in the marine boundary layer contributes to the fine mode AOD and 2) aerosol in clear sky is representative of the above cloud aerosol. As far as the first assumption is concerned, a polluted marine boundary layer with non-negligible black carbon concentrations was observed at times during ORACLES 2016 (ORACLES Science Team, 2017), which would indicate that the proxy ACAOD from MODIS 12-year climatology may be an upper bound of the ACAOD. The synoptic scale of near-constant ACAOD values (see Fig. 1) spans both the marine stratocumulus clouds and neighboring clear sky pixels for given days, leading credence to the second assumption.

Additionally, the filtering of MOD06ACAERO to only apply to retrievals over opaque water clouds (with optical thicknesses greater than 4), may lead to systemic biases in ACAOD. Aerosol embedded within clouds have been shown from spaceborne polarimeter measurements to skew ACAOD retrievals (Deaconu et al., 2017). Although based on different retrieval principles, having aerosol embedded within clouds would likely produce a similar reflectance spectrum in MODIS measurements than aerosol above clouds, leading to biased high retrievals of ACAOD that includes the optical impact of cloud-embedded aerosols.

## 4.4 Vertical profiles of aerosol optical properties
### 4.4.1 Spatial variability in AOD profiles
The vertical distribution of the measured AOD is presented in Fig. 10, with the vast majority representing the ACAOD profiles, and some representing full column profiles. Here, we show a subset of the $AOD_{501}$ profiles divided into northern vs southern geographic regions to compare coastal flights (Fig. 10b & 10d) versus along the further-from-coast routine diagonal (Fig. 10a & 10c). Of particular interest are the considerably high values (>0.5) of $AOD_{501}$ observed in coastal flights at the base of the aerosol plume, compared with similar altitudes (about 2500 m) along the routine diagonal region. The top of the aerosol plume for all these profiles are within the range of 4000 m to 6000 m. In these altitude profiles, which show column AOD of the aerosol only above the aircraft at a given time, a near vertical AOD trace (i.e. no change of AOD with height) denotes a vertical range where the aerosol content is low or its contribution to the total optical depth is marginal, i.e., a gap. Although variability is observed, particularly farther from the coast, such near-vertical lines occur more often and for larger vertical distance along the routine diagonal. Similarly, a negative slope with altitude denotes the presence of aerosol with large impact on the total optical depth. As expected, for the observed profiles, this feature coincides with high concentration of the in situ biomass burning tracer CO (above 200 ppbv) measured from the COMA instrument.

Although generalities can be inferred from these profiles, a high degree of variability is noticeable, especially when contrasting the near-coast profiles versus those along the routine diagonal. This variability is more commonly found in the presence of a gap between cloud and aerosol and the gap's vertical distance. For the coastal flights, the gap's vertical distance ranges from 0 - 2500 m, while for the routine flights it is 0 - 4000 m. As an indicator of the variability of the AOD profiles in these different regions, we observed at 2000 m AOD ranges between 0.17 to 0.6 (0.28 to 0.72) for the southern (northern) profiles along the routine diagonal, and 0.3 to 0.58 (0.35 to 0.93) for the southern (northern) coastal profiles. The vertical thickness of the plume itself is also generally larger in the northern regions (Fig. 10a & 10b), consistent with the climatological understanding of the plume spatial and vertical location (Zuidema et al., 2016).

### 4.4.2 AE vertical dependence
Considering all measurements made during ORACLES 2016 from the P-3, the $AE_{470/865}$ is roughly constant at a median value of 1.75 for the column of aerosol extending from base altitudes ranging between 600 m and 6 km to the top of atmosphere, whereas for column bases below that, the median decreases monotonically to 0.6 (Figure 11).The AE flagged as ACAOD (blue colors, fig. 11) is calculated from individual AOD spectra only for the portions encompassing the entirety of the above cloud aerosol layer. The AE for all data is calculated from AOD spectra representing aerosol above the aircraft altitude, often only partially representing aerosol layers, regardless of whether there are clouds or aerosol in the underlying column. The inclusion of all data permits the quantification of AE at altitudes higher than the highest base altitude of aerosol above cloud layer(s) (which is just shy of 4000 m). The ACAOD $AE_{470/865}$ above 3000 m increases up to 2.1, diverging from $AE_{470/865}$ from all data. Although this may indicate a trend, the low sampling (less than 3 days, denoted by the light color shading) for the ACAOD data at those altitudes may simply be spurious as compared to $AE_{470/865}$ at the

same altitude calculated from all AOD. This larger AE at elevated altitudes for ACAOD seems to indicate that when considering the above cloud AOD only, the ACAOD of aerosol layers with the most elevated bases are likely to be comprised of relatively small particles, especially compared to all data sampled at that same altitude. The relatively consistent $AE_{470/865}$ with altitude is an

indicator of a constant aerosol particle size distribution throughout the vertical layer, above 600 m. Below that, the much smaller average $AE_{470/865}$ is a telltale sign of larger aerosol particles near sea surface, and is reproduced over more than 9 days sampled, even when filtering out the profiles near Walvis Bay (not shown), where there was significant dust. The mean and median are vertically uniform, but there is a larger variability at higher altitudes, especially near 4800 m.

### 4.4.3 Hyperspectral ACAOD profile example

For a singular case, 4STAR's hyperspectral sampling allows analysis of AOD at multiple wavelengths, covering a vast spatial region including vertical flight profiles. Figure 12 shows hyperspectral AODs for the above-aircraft aerosol layer during a selected flight segment on 20

September 2016. This case, sampled near 16.7°S and 8.9°E, has a full-column ACAOD of 0.63 at 501 nm. No gap is observed between cloud top (950m; bottom of profile) and the aerosol layer. There are, however, changes in AOD gradient with altitude, indicating variable aerosol extinction with altitude, likely due to vertical structure of aerosol concentration or type within the full aerosol plume. The top of the aerosol layer extends to 5916 m; there is minimal change in

AOD observed above that altitude. The vertical profile (Fig. 12a) is not always continuous, with some breaks in AOD measurements linked to sampling issues, such as a momentary loss of sun-tracking through a spiral maneuver of the aircraft found at 3500 m of altitude.
AOD measured here has a smoothly varying dependence on wavelength in the ultraviolet to near-infrared range. This vertical profile of AOD shows a mostly constant wavelength

dependence of the AOD at different altitudes (Fig. 12b). In addition to the AOD, we included total optical depth, which includes the contributions of strongly absorbing gas components (water vapor, oxygen A-band) in shaded wavelength regions. The AOD spectra at different altitude (Fig.12b) are seen to be mostly smoothly varying, except for locations of low signal to noise of the 4STAR's detectors, such as the longest wavelength region near 1600 nm, and at

wavelength regions near 430 nm, where a slight 'bump' over the smoothly varying spectra are observed and likely linked to signal issues of the detectors.

Figure 13 shows profiles of ACAOD at specific wavelengths (Fig. 13a), as well as the $AE_{470/865}$ as an indicator of above aircraft aerosol particle size (Fig. 13b). The $AE_{470/865}$ does not change

significantly from 1.75 for altitudes up to 4500 m, above which it is reduced down to 1.25 corresponding with low AOD (<0.05). The aerosol extinction coefficient can also be derived for the AOD vertical profile (Fig. 13c) by using the differential of AOD with respect to altitude change with a smoothing of 50 seconds (similarly to Shinozuka et al., 2013). This extinction coefficient compares well to the in situ extinction coefficient (Fig 13d), derived using the

HiGEAR's nephelometers for scattering coefficient adjusted to ambient relative humidity and the absorption coefficient of dry particles measured using the PSAP. We also see that regions of high extinction coefficient track well with elevated CO concentration for this profile (Fig. 13e). Slight deviation between the extinction coefficient calculated from 4STAR AOD and in situ measurements are likely linked to differing RH dependence of the aerosol particles, and its

adjustments, particularly where there is variability of the ambient RH, or when there is different instrumental representation of the RH scattering absorption. The relative humidity for this profile is between 10% and 80% within the aerosol layers (Fig. 13f), with the majority of the profile near 20% RH.

## 4.5 AOD distance to cloud

The vertical profiles of AOD showcase the large variability in the gap size and location along the atmospheric column (Fig. 10). The ACAOD flag, described in Section 3.1, allows assessment of the frequency of cases where there is and is not a gap between aerosol layer and cloud, (Fig.
2b & 2c), though is not able to identify more complex scenes with a gap within aerosol layers. During any one profile, the vertical extent of the continuous measurements flagged as ACAOD quantifies the gap between cloud top and aerosol layer bottom. For cases where this vertical extent is near 0 m (within an uncertainty of 60 m), it is said that the profile has no gap between aerosol and cloud. Unlike previous studies from spaceborne lidars (Devasthale and Thomas,
2011; Rajapakshe et al., 2017), we found that within the entire region sampled by the NASA P-3 the gap does not linearly decrease towards the west in a near-monotonic fashion (Fig. 14). Figure 14a shows the meridionally averaged gap extent for all the samples, convolving the temporal and latitudinal variations. The smallest gap extent is observed at longitudes westward of 2.0°E, similarly to CALIOP measurements (not shown; Wood et al., In prep.), but may be
biased due to the low number of days sampled (only a maximum of 3 days, with 6 different profiles) resulting in a relatively large impact of the meteorological state comparatively to the driving impact of the climatology. The largest average gap is not nearest to coast but rather midway in this sampling region at about 7.5°E, and is observed over 5 non-consecutive days spanning 8/31 to 9/20, with gaps larger than 1km observed on 9/06 at 18.2°S, on 9/10 at
17.8°S, and on 9/14 at 16.1°S to 17.7°S. Similarly, a local maximum in gap extent near 7.5°E was described by Rajapakshe et al. (2017) using observed in nighttime CATS and CALIOP measurements. Nearer to coast, between 8.5°E and 11.5°E, there is a region of smaller to near zero gap extent, with median extents below 500 m. Combined together in larger longitude spans with higher number of samples (Fig. 14b, 14c, and 14d), omitting the profiles taken over land
during take-off and landing at 14.5°E, the mean of the gap extent distribution peaks between 5°-10°E. Another way to view this distribution's dependence with longitude is the proportion of the total profiles or cases that have a gap of less than 60 m (near zero for this analysis), or through the larger distance defined by McGill et al. (2015) as Clouds embedded within an Aerosol Layer (CEAL; 360 m), denoted by the dark and light gold colors in Fig. 14. We see a region where 0%
of the 3 days (4 profiles) measured a near zero gap extent at 5.5°E, and 0% of the 3 days (16 profiles) are considered CEAL cases at 5.5°E to 7.5°E. The peak of the cases that have no gap or CEAL occur at the westernmost edge, with a secondary peak between 8.5°E to 11.5°E. For all measurements, the proportion of CEAL cases is observed here at 48%, a statistically significant lower value (p-value of 0.027) than reported for a larger region sampled with CATS
(60%) by Rajapakshe et al. (2017).

The direct radiative effect of aerosol above clouds is not likely to be modified significantly whether the aerosol is touching or not the top of the cloud, but rather the modulation of inherent aerosol and cloud properties. The direct aerosol radiative effect varies by only 1% - 3% when

considering changes of height above cloud of back carbon aerosol layer (Zarzycki and Bond, 2010). Alternatively, for the indirect cloud-aerosol interactions, we have observed aerosol layers touching the top of the clouds. We've observed more direct contact between clouds and aerosol by up to 12% for CATS as reported by McGill et al. (2015), and potentially by more than 40% for

CALIPSO as compared to Devasthale and Thomas (2011), this increasing the potential of a larger indirect effect. Albeit, touching of the aerosol and cloud is not always the best indicator of potential aerosol-cloud interactions for indirect effects, especially when considering that there may have been past interactions between a specific cloud and aerosol layer (e.g., Diamond et al., 2018). The exact representativeness of these results, including the aerosol layer vertical

distribution, from airborne sampling to the natural world are investigated in future studies (e.g., Shinozuka et al., Submitted to ACP). There is likely a large inter-annual variability and geographical sampling variations in the SEA, which could skew the comparison between airborne and satellite sampling.

## 5 Summary and Discussion

During the ORACLES 2016 campaign, the NASA P-3 sampled aerosol above marine stratocumulus clouds in the South-East Atlantic during the month of September, coinciding with the peak of the biomass burning season in Sub-Saharan Africa. The 4STAR instrument, on board the P-3, sampled the AOD from a range of flight altitudes, a portion of which is defined as ACAOD. The ACAOD is presented here in terms of distribution of its magnitude, spatial

dependence, vertical variability, and spectral dependence.

For all measured spectral AOD during September 2016, different statistics (mean, median, and standard deviation) are calculated by two methods, summarized in Table 1: first by averaging all measurements equally, and second, utilizing spatial binning before averaging to assess the influence of highly-sampled regions. By calculating the mean, median, and standard deviation

from all measurements, we inherently give more weight to regions most often sampled during the field campaign (specifically the routine flight paths), whereas the spatial binning of these statistics represents a more evenly spatially-weighted representation of the measured values. Here we see that the mean spatially binned ACAOD is higher than from all measurements, indicating that we disproportionally sampled low ACAOD regions, similarly for the Total AOD,

and ACAOD uncertainty. The spatially binned AE is smaller than its all measurement counterpart, showing that our sampling locations and focus was biased high for smaller aerosol particles in comparison to a more evenly spatial distribution.

Observed variations in AOD and AE during the sampling period are significant, from changes in

spatial patterns to changes in vertical profiles. The northern region near coast sees the largest measured optical depth, as observed in the spatial pattern of the ACAOD. This is also where 12 years of MODIS AOD sampling shows the most optically thick aerosol plume. Along the diagonal flight path, measured during routine flights from the NASA P-3, the lowest ACAOD is observed at the southern end, with the largest variability of ACAOD midway, linked to the

latitudinal movement of the aerosol plume's southern edge. This region of high ACAOD variability coincides with high variability of the $AE_{470/865}$ derived from the ACAOD spectral

dependence. This coincident high variability indicates that we sampled a mixture of aerosol particle populations comprised of a majority of small particles from the optically thicker biomass burning plume and a minority of aerosol particle with larger variability in aerosol size or composition near the southern edge of the climatological plume. Looking at the ensemble of the region, Table 1 shows that for the full column AOD, the $AE_{470/865}$ is lower than the AE from ACAOD. This is more evident when considering the spatially binned AE from full column AOD vs. ACAOD, which are well outside one standard deviation from their respective means. This notion is also supported by the vertical profile of AE (Fig. 11) which indicates the presence of large aerosol particles, potentially marine aerosol embedded within the lower boundary layer, only present when considering the full column AOD.

When comparing to satellite measurements and long term AOD measurements in the region, the measured ACAOD is lower than both coincident MOD06ACAERO retrievals and the long-term fine mode MODIS clear-sky AOD average over the region. 4STAR systematically reports lower ACAOD by 0.05 – 0.08 than MOD06ACAERO, even when considering only the days sampled by the aircraft. The ACAOD from 4STAR also has a peak closer to shore, and more south than the MODIS AOD climatology mean and median (both fine and coarse mode), with differences near coast between 4STAR ACAOD measurements and MOD06ACAERO retrievals. Differences between 4STAR ACAOD and the MOD06ACAERO subsampled for the same day are possibly linked with daily airmass movement and underlying cloud diurnal cycle, especially when there is a mismatch between MODIS overpass times and aircraft sampling times. The subsampled MOD06ACAERO is more similar to the August mean average than the September average, which can partially explain the sampling representativeness, and therefore some differences, between 4STAR ACAOD and September climatology built from MODIS measurements.

The regions where the largest divergence between MOD06ACAERO and 4STAR ACAOD coincides with the largest variability in AE (near 7°E), and likely indicates a link between aerosol properties and the accuracy of MOD06ACAERO. Complicating factors for satellite retrievals in this region may be linked to the occurrence of mid-level clouds topping the aerosol layer, which have been observed in this region and have also been reported, in the form of elevated RH, to occur over a longer time sample from satellite and sounding observations by Adebiyi et al., (2015). Differences between MOD06ACAERO and 4STAR ACAOD may also be attributable to satellite retrieval sensitivities to aerosol embedded within clouds, although these differences do not seem to correlate with the gap extent. Embedded aerosol within clouds is still possible through the inclusion of marine boundary layer aerosols mixing upwards in clouds, or past mixing of above-cloud aerosol into underlying clouds (Diamond et al., 2018). Other possible sources of differences may be the underlying selection of aerosol model (aerosol single scattering albedo, asymmetry parameter, etc.) in the MODIS ACAOD retrieval or the cloud mask applied (i.e., only using cloud of optical thickness 4 and above). Here we found a smaller $AE_{470/865}$ (mean: 1.71), than what is defined in the aerosol model within the MO06ACAERO retrieval (~2.0 when the AOD at 550 nm is 0.5 from Levy et al., 2007, with an AOD dependence), which may suggest the underlying aerosol model needs refinement.

Differences in vertical AOD profiles are indicative of the variability of the altitude and magnitude of the aerosol plume. We have observed distinct AOD profiles along the routine diagonal and for coastal flights. Coastal flights typically had larger AOD at high altitude (averaging to 0.51 at 2500 m altitude) as compared to flight along the routine diagonal (averaging to 0.38 at 2500 m altitude). The vertical extent where the AOD does not change significantly, here used to indicate a gap between aerosol and cloud, spans a larger distance further from coast than near coast (0-4000 m far-from-coast, 0-2500 m near-coast). A strong decrease in AOD with increasing altitude coincides with locations of high concentrations of CO, a tracer of biomass burning. The derived extinction coefficient from 4STAR AOD profiles and in situ measurements appear to match very well for one example shown. In the vertical domain, the AOD is observed to be spectrally smooth with $AE_{470/865}$ nearly vertically constant for the majority of the measurements, only significantly decreasing near surface. The gap vertical extent calculated from 4STAR data, in conjunction with in situ measurements of scattering coefficient and cloud drop concentration, appears to have a more complex dependence with longitude than was initially expected from CALIOP space-borne observations. Visual observations from the NASA P-3 flights corroborate previous observations of clear air slots, and their inherent variability. There is a prevalence of near zero gap extent, while the largest gaps extents are observed not close to coast, as expected, but off-shore near 7° E. We have also observed a lower proportion of cases where the aerosol layer is near the cloud top as compared to previous studies (48% of CEAL instead of the 60% reported using CATS by Rajapakshe et al. (2017).

From these airborne measurements, we have seen that the ACAOD is lower than expected from subsampled MODIS satellite retrievals (MOD06ACAERO) during the measurement period (by 0.05-0.08) and from a 12-year climatology (by 0.04). We have also observed the largest variability in aerosol optical properties (ACAOD and AE), at the southern edge of the climatological aerosol plume for September. The vertical dependence of the ACAOD was highly variable, even for the same regions, with aerosol layer tops ranging from 4000 m to 6000 m, while their bottoms were from 400 m to 4000 m. We observed that the extent of the aerosol-cloud gap peaked at a longitude of 7.5°E, unlike the expectation of a gradual decrease of this gap as the aerosol plume moves westward, further from coast.

## A Appendix: Description of 4STAR AOD data quality

AOD sampled by 4STAR is subject to various sources of measurement uncertainty (stability of calibration coefficients, sun tracking accuracy, dark count stability, air mass calculations, Rayleigh scattering subtraction, gas absorption impact, and diffuse light contributions; see Appendix A in Shinozuka et al., 2013). In addition to uncertainty sources described by Shinozuka et al., (2013), we include for ORACLES 2016 4STAR AOD: 1) the impact of changes of calibration linked to changing spectrometer throughput during the field mission, 2) impact of in-flight window deposition, and 3) impact of angular response to radiometric calibration of the 4STAR head. These corrections are processed within the 4STAR's open source processing code (4STAR Team, 2018).

## A.1 4STAR calibration and performance

To calculate AODs from 4STAR, we obtain a radiometric calibration in terms of the inferred signal that would be observed by 4STAR at the top of the atmosphere using a refined Langley extrapolation method based on the Beer-Lambert law (Schmid and Wehrli, 1995), used by Shinozuka et al., (2013). To reduce the potential for calibration bias, we use a collection of calibrations from refined Langley extrapolations near sunrise and sunset taken from airborne measurements and from the high-altitude Mauna Loa Observatory (MLO) in Hawaii. The airborne calibrations (5 total) were executed during high altitude portions of flights (including the transit flights), with low calculated AOD (below 0.05 at 501 nm), and an airmass change of greater than 2. The Langley extrapolations from MLO were taken weeks before (pre-deployment) and after (post-deployment) the observation campaign, under minimally-polluted conditions with a spread of airmass factor from 1.8 to 12.  Using similar metrics to those described by Shinozuka et al. (2013), the relative standard deviation of the calibration derived from 6 Langley extrapolations during pre-deployment MLO is 0.63% (0.17%) at 501 nm (1040 nm). For post-deployment MLO, this relative standard deviation calculated from 4 Langley extrapolations is 1.2% (0.39%) at 501 nm (1040 nm). For all in-flight Langley extrapolation, we obtained a relative standard deviation of 1.1% (0.91%) at 501 nm (1040 nm), deviating from the post-deployment MLO by 0.99% higher at 501 nm and 0.56% lower at 1040 nm. The calibration from the post-deployment MLO Langley extrapolations shows a decrease of 2.9% (an equivalent maximum AOD of 0.029 when sun is overhead) at 501 nm and an increase of 0.2% (equivalent to 0.002 AOD) at 1040 nm as compared to pre-deployment MLO. This variation between the pre- and post-deployment MLO calibration is attributed to disconnection of the fiber optic linking the 4STAR head and the spectrometers during the time between the MLO pre-deployment calibration and the ORACLES deployment. Subsequent disconnections did not occur. Because of this disconnect, we did not use the pre-deployment MLO calibrations for ORACLES data, but its repeatability helps describe the instrument's precision over multiple weeks, for an unaltered instrument condition.

During ORACLES, the AOD derived from 4STAR measurements were sensitive to relative humidity variations of the spectrometers, when failure of the humidity control occurred (desiccant was depleted). To mitigate these effects, we incorporate another calibration from AOD measured under high altitude, near solar noon, low aerosol loading conditions when 4STAR was effectively sampling the stratospheric AOD contribution, and was subjected to different spectrometer humidity. A set of new calibrations was obtained from the average of Langley extrapolation obtained during post-deployment MLO, in-flight Langley extrapolations, and from calibrations derived from matching a reference stratospheric AOD spectrum to high altitude high sun measurements. The reference stratospheric AOD spectrum is obtained from the lowest AOD measured at the AERONET (Holben et al., 1998) Bonanza, Namibia, site (an altitude of 1.3 km) over the course of 3 months, which was found to be 0.016 at 501 nm, and then a log-log second-order polynomial fit (e.g., Shinozuka et al., 2013) was used to interpolate the reference AOD spectrum to the wavelengths sampled by 4STAR. From this method, a total of 7 sets of calibrations (described within the archived 4STAR AOD data; ORACLES Science Team, 2017) were applied to 4STAR, separating periods of varying relative humidity of the enclosure containing the spectrometers. The relative standard deviation of all these calibrations

is 0.83% (1.12%) at 501 nm (1040 nm). Similar performance from 4STAR has been observed in previous field campaigns (e.g., Shinozuka et al., 2013), where extensive comparisons to ground based AERONET stations resulted in a root-mean-square difference of 0.01 for wavelengths between 501 nm and 1020 nm, 0.02 at 380 and 1640 nm, and 0.03 at 440 nm.

## A.2 4STAR corrections and uncertainty

Accurate 4STAR measurements of AOD require corrections for some instrument artifacts and impact of light absorption by trace gases. Corrections related to light transmission variations due to angular variability of the fiber optic rotating joint (FORJ), due to deposition of material on the
outside window of 4STAR's sun barrel, and finally atmospheric trace gases contribution to AOD estimates.

Light transmission variability due to the FORJ is corrected using the azimuthal position of the 4STAR sun-tracking head in relation to the plane's axis. This azimuthal dependence is
measured in between each flight by a full rotation in each direction while staring at a stable light source (a light emitting diode that has less than 0.1% variation in radiance during the time of the test). The variations have a near sinusoidal shape with features departing from the mean by no more than 1.4% and are repeatable in between each measurement (within 0.2% over the course of the field mission), with the largest features not moving by more than 30 degrees.

The impact of window deposition on the transmission of 4STAR's sun barrel is quantified by measuring the change in signal from a stable light source before versus after cleaning the window, and is performed after each flight. We attributed any window deposition observed to discrete events during flight, notably during low-level near water flight segments or during cloud
insertions. The uncertainty of the AOD surrounding these events (within +/- 20 minutes) have been increased to the magnitude of the window deposition's optical depth, and by 30% of the corrected magnitude for the rest of the flight, producing a step-change in the AOD uncertainty. The impacts of these events were quantified by the change in high altitude AOD before and after the low-level segments. Differences of larger than 2% but not more than 4.5% occurred in
4 of the 15 research flights and have been accounted for, both the magnitude of the AOD and its related uncertainty, using the above described method.

AOD is influenced by trace gas absorption in the entire column in distinct wavelength regions. We correct the influence of trace gas ($NO_2$, $CO_2$, $O_3$, $O_2$-$O_2$, $CH_4$) by convolving their retrieved
vertical column gas abundance and profile with their spectral absorption coefficients (Segal Rozenhaimer et al., 2014). This result in an optical depth contribution from these gases (typically very minor) which is then subtracted from the AOD spectrum.

## Data availability

All ORACLES-2016 in situ data used in this study are publicly available at https://doi.org/10.5067/Suborbital/ORACLES/P3/2016_V1 (ORACLES Science Team, 2017).

This is a fixed-revision subset of the entire ORACLES mission dataset. It contains only the file revisions that were available on 15 June 2018.

## Author Contributions

SL and JR conceived the study. JR acquisitioned the funding. SL and KP analyzed the data with
help from CF, ML, MSR, YS, and SGH. YS helped in curating the data from ORACLES. SL, CF, KP, MK, MSR, YS, JP, SGH, SF, JSG, MD, and RW collected data on board the NASA P3, while AERONET data was collected by BH, PF, SP, GMK, MG, and AN. KM and RW provided satellite data and analysis. SD and RPD provided engineering support for 4STAR. SL wrote the paper with reviews from all authors.

## Acknowledgments

The authors wish to acknowledge all of the ORACLES science team and the NASA P-3 flight and maintenance crew for the successful deployment. ORACLES is funded by NASA Earth Venture Suborbital-2 grant NNH13ZDA001N-EVS2. The Henties Bay and Gobabeb AERONET stations are maintained by the French Centre National de la Recherche Scientifique (CNRS)
and the South African National Research Foundation (NRF) through the "Groupement de Recherche Internationale Atmospheric Research in southern Africa and the Indian Ocean" (GDRI-ARSAIO), the "Projet International de Coopération Scientifique" (PICS) "Long-term observations of aerosol properties in Southern Africa" (contract n. 260888), and the Partenariats Hubert Curien (PHC) PROTEA of the French Ministry of Foreign Affairs and International
Development (contract numbers 33913SF and 38255ZE).

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

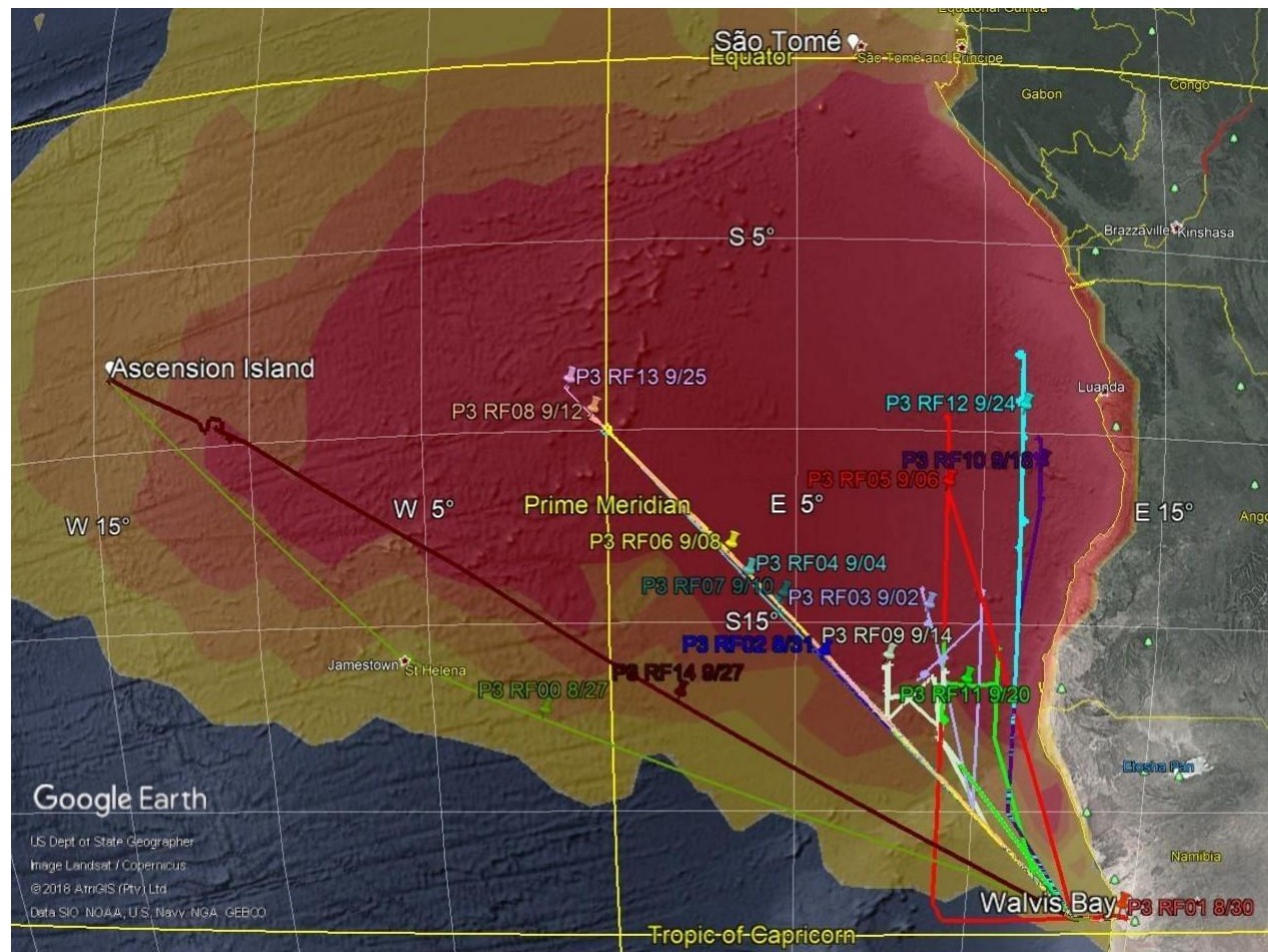

*Figure 1 – Map of the South-East Atlantic (SEA) region with flight paths from the NASA P-3B during ORACLES deployment of 2016. Climatological aerosol optical depth from MODIS for September (2001-2013) is overlaid as colored shaded contours (yellow shading represents AOD of 0.25, with deep red shading for 0.5 (adapted from Zuidema et al., 2016)).*

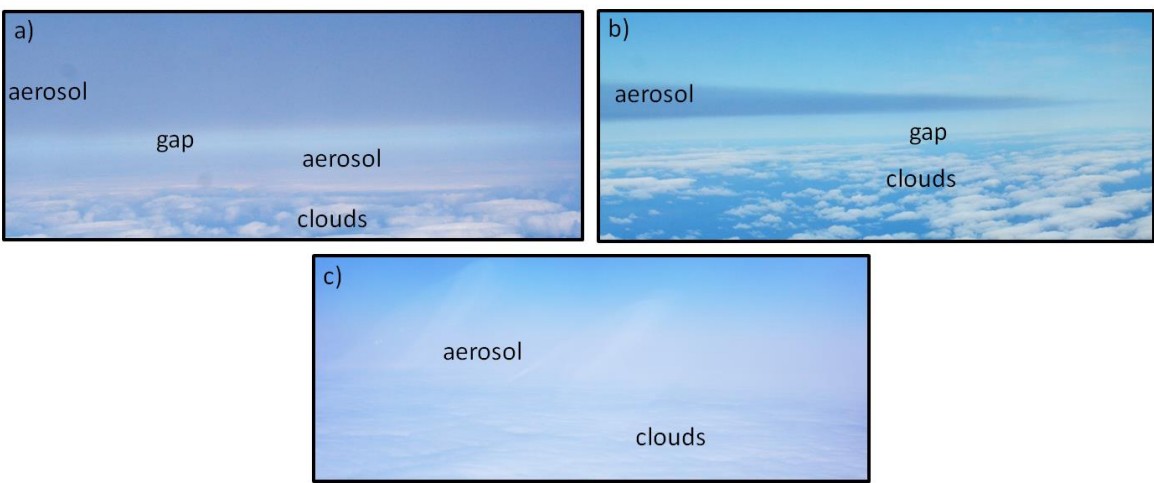

*Figure 2 - Photographs taken from the P-3 of (a) a gap between two aerosol layers, (b) a gap between an aerosol layer and cloud, and (c) no gap between aerosol and cloud.*

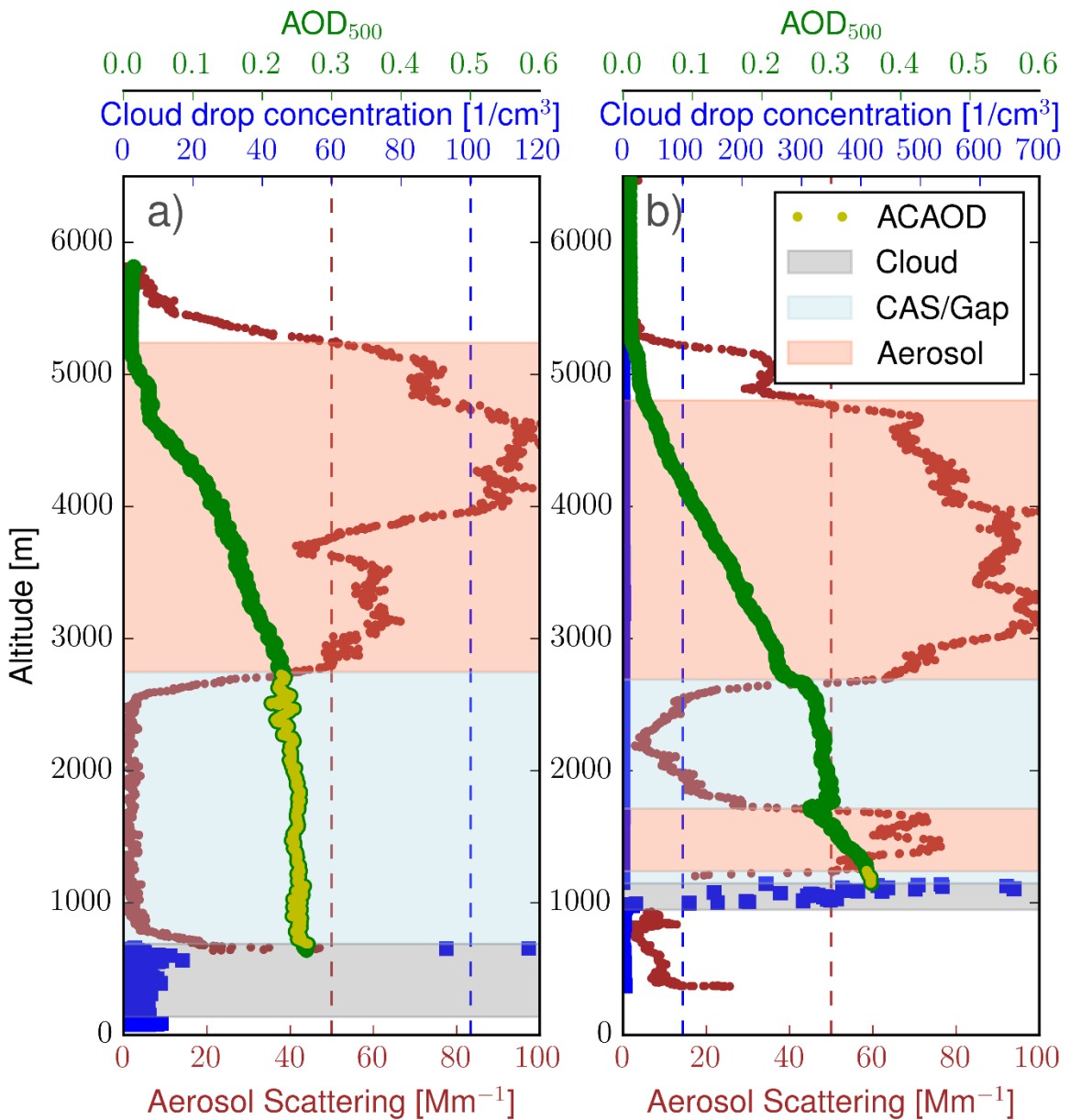

*Figure 3 - Examples of profiles of cloud drop concentration from PDI, aerosol scattering (at 550 nm) from HiGEAR's nephelometer, and AOD measurements used to evaluate the ACAOD portion of the total AOD column taken from flight on 2016-09-12. a) case from 18.6°S, 8.6°E where there is a gap (light blue shading) between cloud top (grey shading) and an aerosol layer (light red shading). The yellow markers within the green AOD profile denotes the vertical portion of the flight representing the ACAOD. b) case from 10.2°S, 0.2°E with a near zero separation between cloud top and aerosol layer, but with an embedded gap within the aerosol layer. For this case, only the AOD directly above cloud is considered ACAOD.*

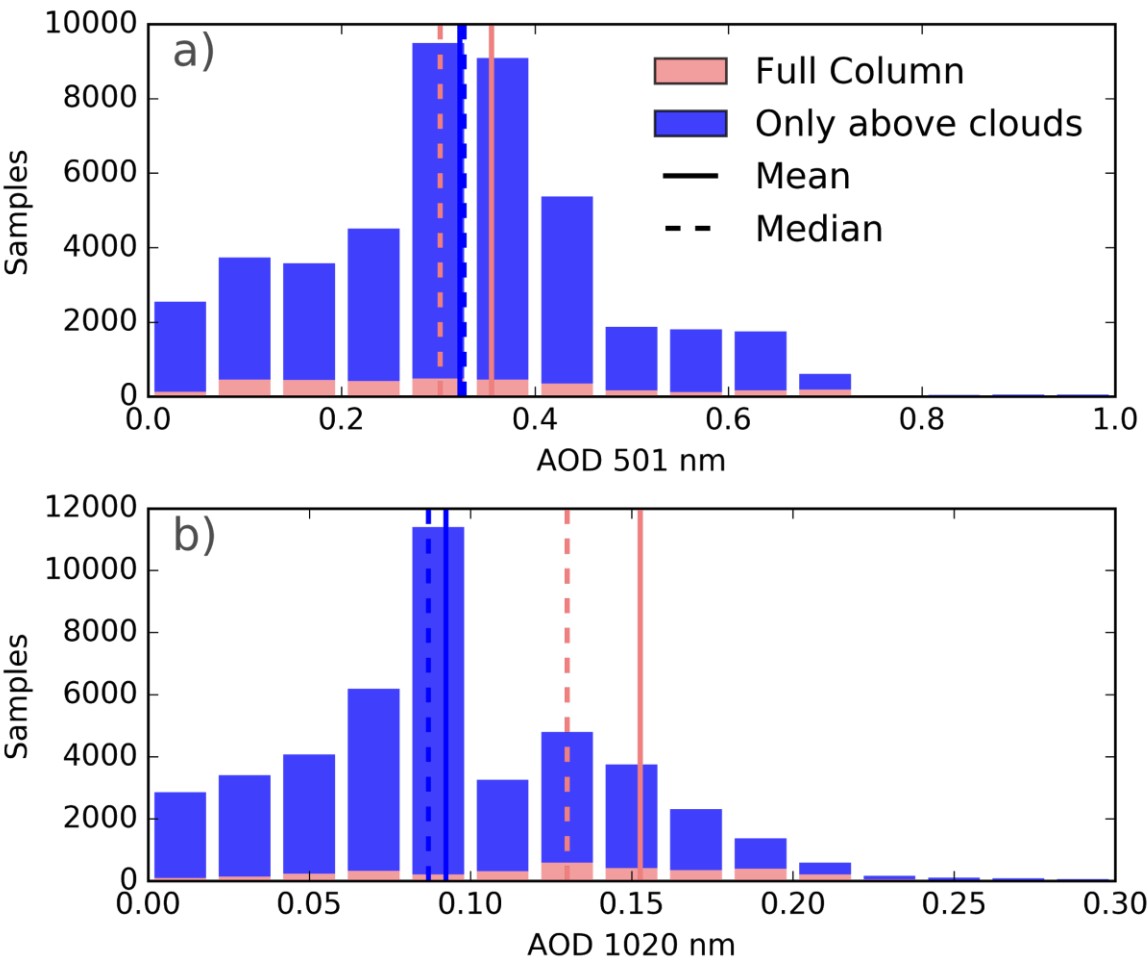

*Figure 4 - Histograms of above-cloud (blue) and full-column (pink) AOD sampled by 4STAR at (a) 501 nm and (b) 1020 nm. 'Full column' denotes sampling below an altitude of 0.6 km where no cloud is between 4STAR and the sun (N=3,388), while 'Only above clouds' denotes the AOD flagged to be only above clouds (see Sect. 3.1, N=41,189). Vertical solid lines denote the mean of the distribution (colored accordingly), while dashed vertical lines denote the median.*

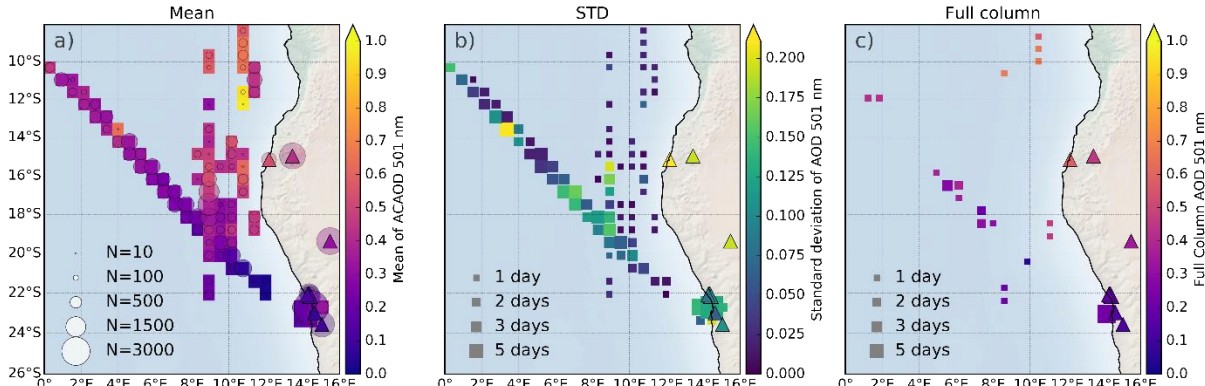

*Figure 5 - (a): Map of mean ACAOD$_{501}$ from all P-3 flights spatially binned during ORACLES 2016 deployment period. The triangles indicate the location of the ground based AERONET stations, colored by their average full column fine mode AOD$_{501}$. The overlaid circle size denotes the number of individual*
5     *samples within that bin. (b): The standard deviation of ACAOD$_{501}$ within each bin with the size of the squares denoting the number of days sampled within each bin. The legend in the bottom left of the panel denotes the different sizes of the square symbol relating to the number of sampled days in each bin. The triangles indicate the standard deviation of the fine mode AOD measured by the ground based AERONET stations from north to south: Lubango, Namibe, DRAGON network at Henties Bay, Walvis Bay Airport,*
10     *and Gobabeb. (c): The mean full column AOD$_{501}$ measurements and their location, with size of the square denoting the number of days sampled. The associated AERONET locations in triangle are for the total (fine + coarse mode) AOD.*

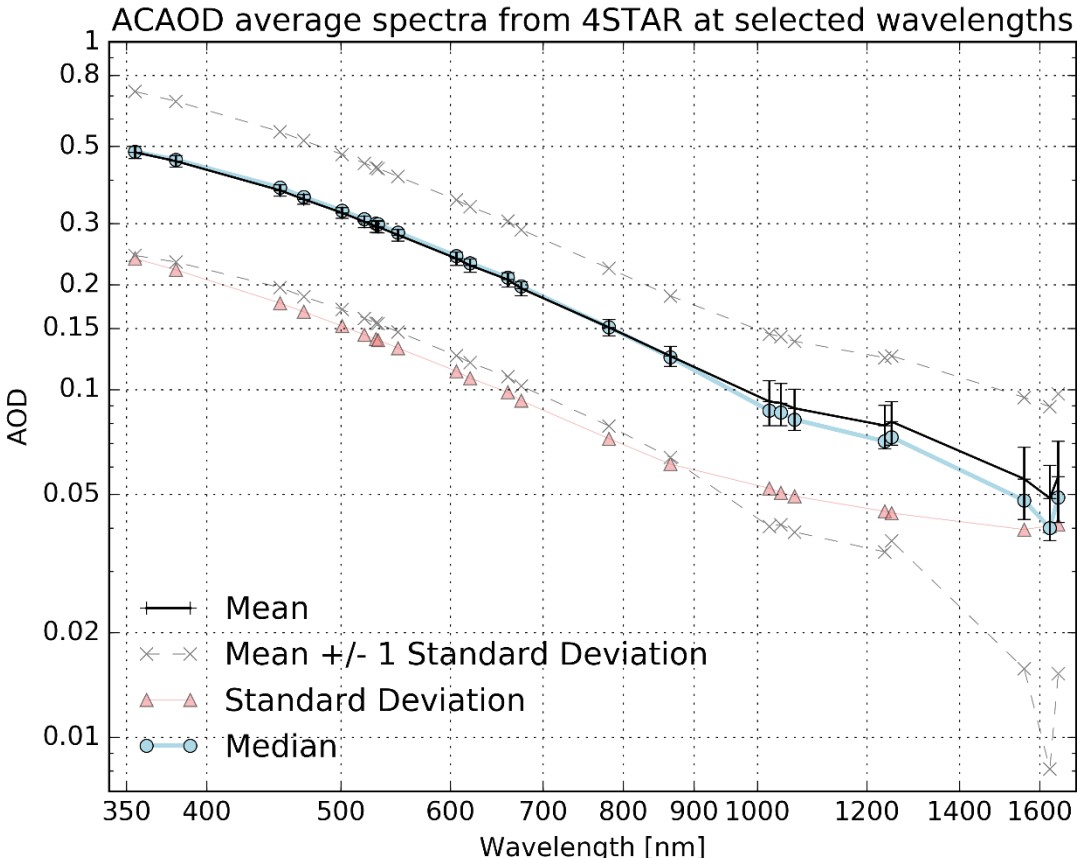

*Figure 6 - ACAOD spectra representing the mean, median, and standard deviation of measurements by 4STAR for selected wavelengths, which have minimal influence of gas absorption and high signal to noise ratio. The mean measured ACAOD at each wavelength is shown in black, along with its mean uncertainty (as error bars in black), median in blue circles, and the range of 1 standard deviation surrounding the mean for all the measured ACAOD (grey dashed lines). The magnitude of the standard deviation is also included, denoted by a thin pink line with triangles.*

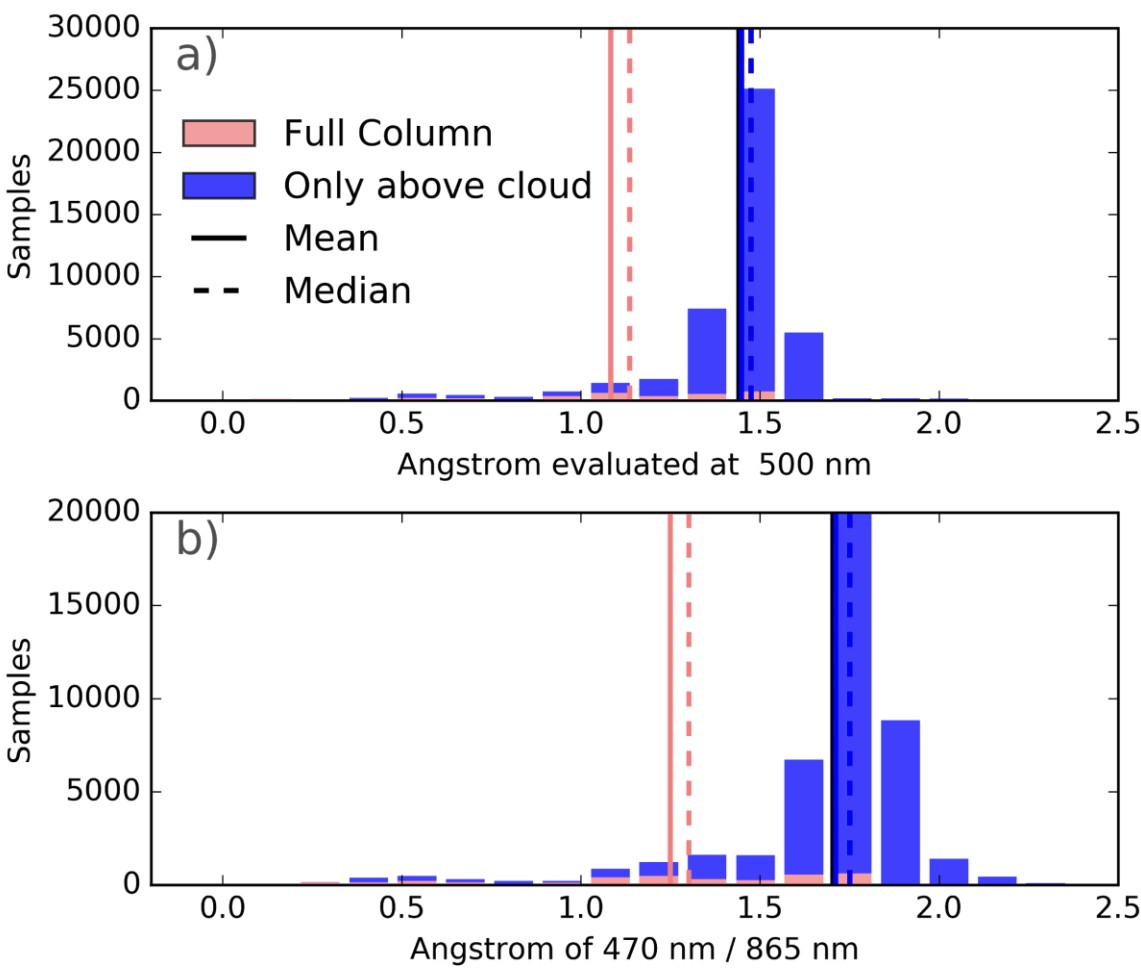

*Figure 7 - Histograms of Ångström exponent (AE) calculated from (a) a polynomial fit of AOD sampled by 4STAR evaluated at 500 nm and (b) using the 2-wavelength ratio (470 nm and 865 nm) in log-normal space, for the full column AOD (pink) and the ACAOD (blue).*

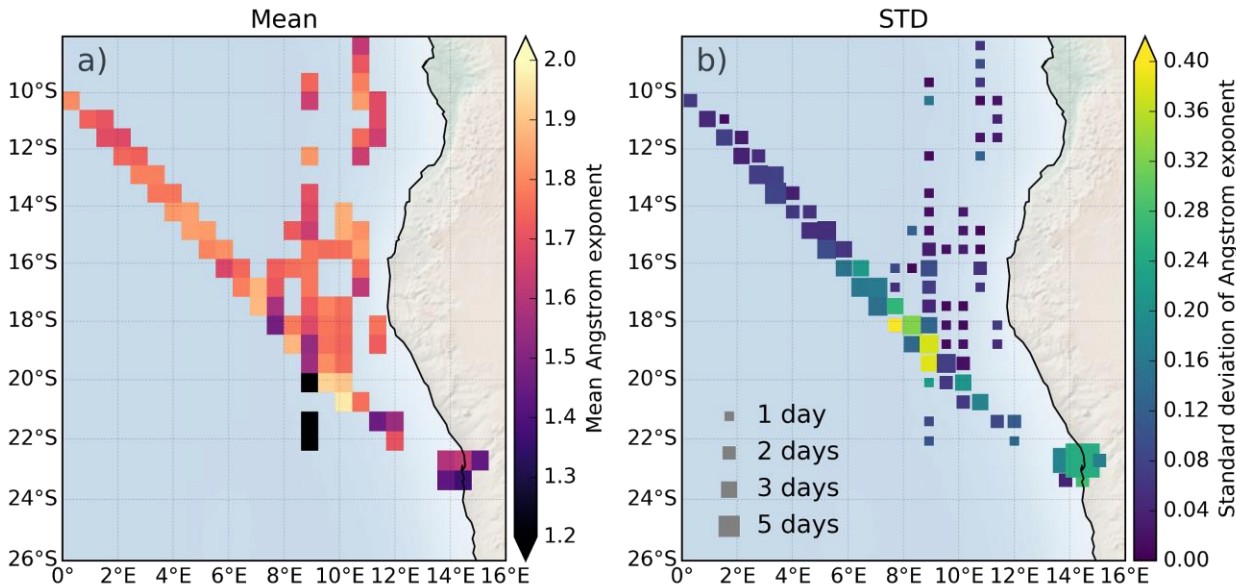

*Figure 8 - Map of mean AE$_{470/865}$ derived from AOD spectra of aerosols above clouds calculated from two wavelengths (470/865 nm) (a), and the standard deviation of the AE$_{470/865}$ (b), where the size of the squares represents the number of sampling days used to build the statistics within each gridded bins, nearly the same number as shown in Fig. 5a.*

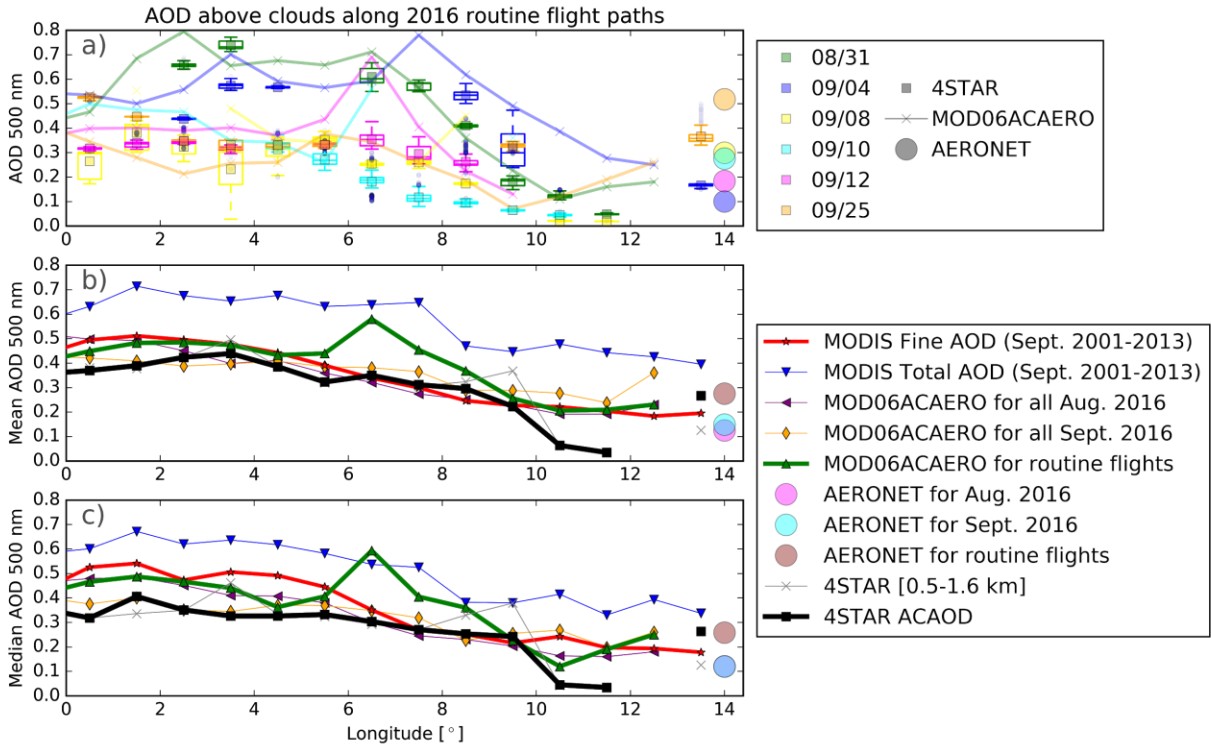

*Figure 9 - ACAOD at 501 nm along the diagonal routine flight path (13°E 23°S to 0°E 10°S) for ORACLES 2016 compared to a MODIS climatology, MOD06ACAERO (Aerosol Above Cloud retrieved from MODIS satellites (Meyer et al., 2015)) retrievals as a function of longitude, and nearby ground based AERONET*
5 *fine mode AOD. (a) The 4STAR ACAOD sampled during the days when the NASA P-3 followed the routine flight path and its equivalent retrievals from MOD06ACAERO. The 4STAR ACAOD is represented by box whisker plots, for binned longitudes, whereas the MODIS AAC is represented by its mean value within a longitude by an 'x' and connecting line. The AERONET fine mode AOD measured from DRAGON at Henties Bay, Namibia for the same days are presented in the far right as circles. (b) The mean of the*
10 *ACAOD sampled over the days listed in the top panel for 4STAR and MOD06ACAERO compared to other retrieved measurements over a longer time period. The monthly mean MOD06ACAERO for August and September 2016, along with the clear sky mean total and fine mode AOD from MODIS from September averaged over the years 2001 - 2013. The mean AOD from 4STAR sampled within the altitude range of 0.5 - 1.6 km. (c) Median ACAOD instead of mean.*

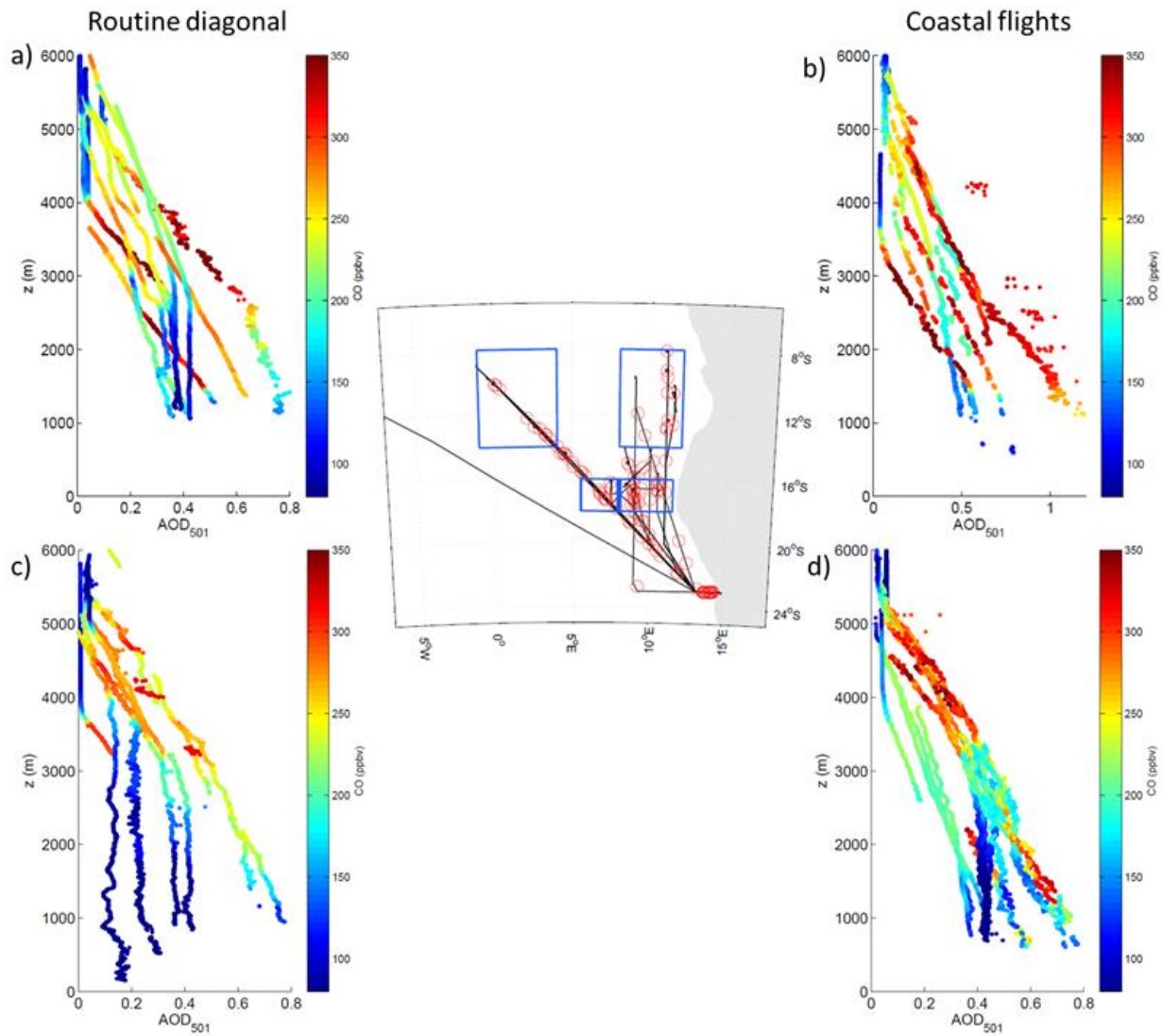

*Figure 10 - A subset of AOD at 501 nm vertical profiles along the routine diagonal (left) and near the African coast (right) at the northernmost edge of the flight tracks (top; 8 to 14S) and near the bottom edge of the plume (bottom; 16 to 18S). Note that only a subset of profiles, roughly equal for each area, are shown for clarity of interpretation, though the middle-latitude profiles generally exhibit features of both latitude bins shown. Color indicates the CO concentration of the ambient airmass, measured by the in situ COMA instrument. The aerosol-cloud vertical gap is most prominent farther from the coast, as indicated by altitudes where low CO values are measured simultaneously with a low vertical gradient in AOD. Flights near the coast show more variability, and fewer cases of an unpolluted gap above cloud (greater low-level CO and stronger gradient of AOD with altitude), although each condition is seen within both regions. The central map shows the location of the subsets overlaid by all flight paths from ORACLES-2016 (black lines), and all P-3 aircraft profiles (red circles).*

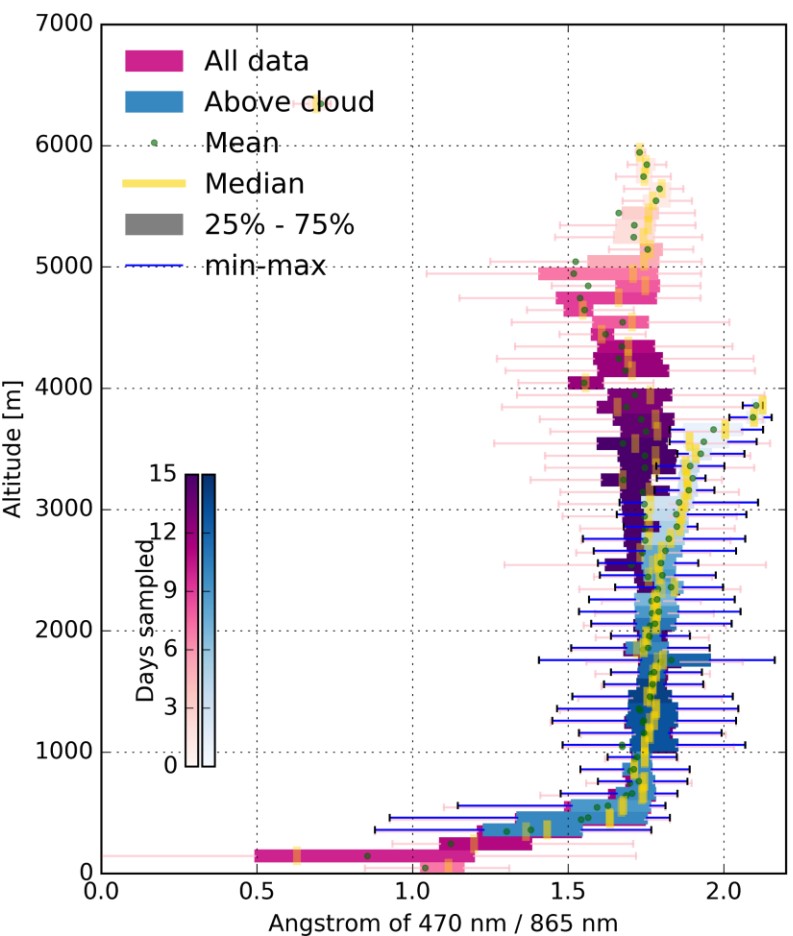

*Figure 11 - Binned vertical profile of $AE_{470/865}$ for all measured AOD greater than 0.1, including all data (red-purple colors) and aerosol flagged as representing ACAOD (blue colors). These represent the $AE_{470/865}$ calculated from all AOD spectra representing the aerosol above that altitude, and binned by 100 meters. The mean of each binned vertical population is represented by the green circle, median by the gold vertical line, the thick horizontal line represents the span of $AE_{470/865}$ from the 25th to the 75th percentile, while the range is denoted by the span of the thin blue (or pink) line. The shading of each box-and-whisker plot denote the amount of days sampled within this altitude bin, linked to the color bars on the left side.*

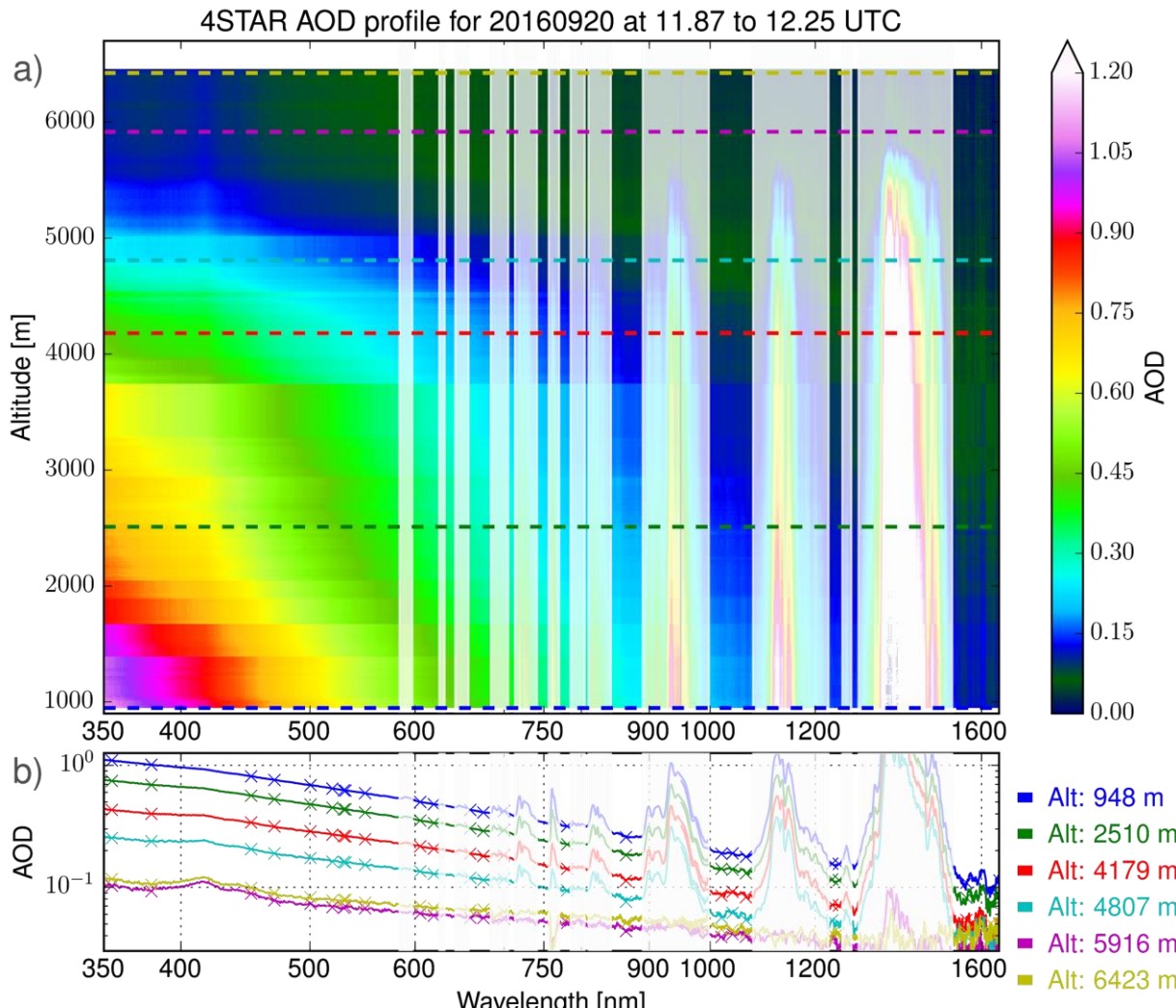

Figure 12 - Hyperspectral AOD profile from 20 September 2018, from a square spiral at 11:52 to 12:15 UTC. a) shows the AOD as the color (linked to the color bar at the far right) continuously and as a function of wavelength and altitude. The shaded regions denote where strong gas absorbers, namely water vapor and oxygen impact the spectra. b) hyperspectral AOD at select altitudes, denoted by the dashed lines in a). The 'x' symbols denote the particular wavelengths at which the AOD is available in the ORACLES data archive, matching some wavelengths used by other instruments, and which the AOD is of highest confidence.

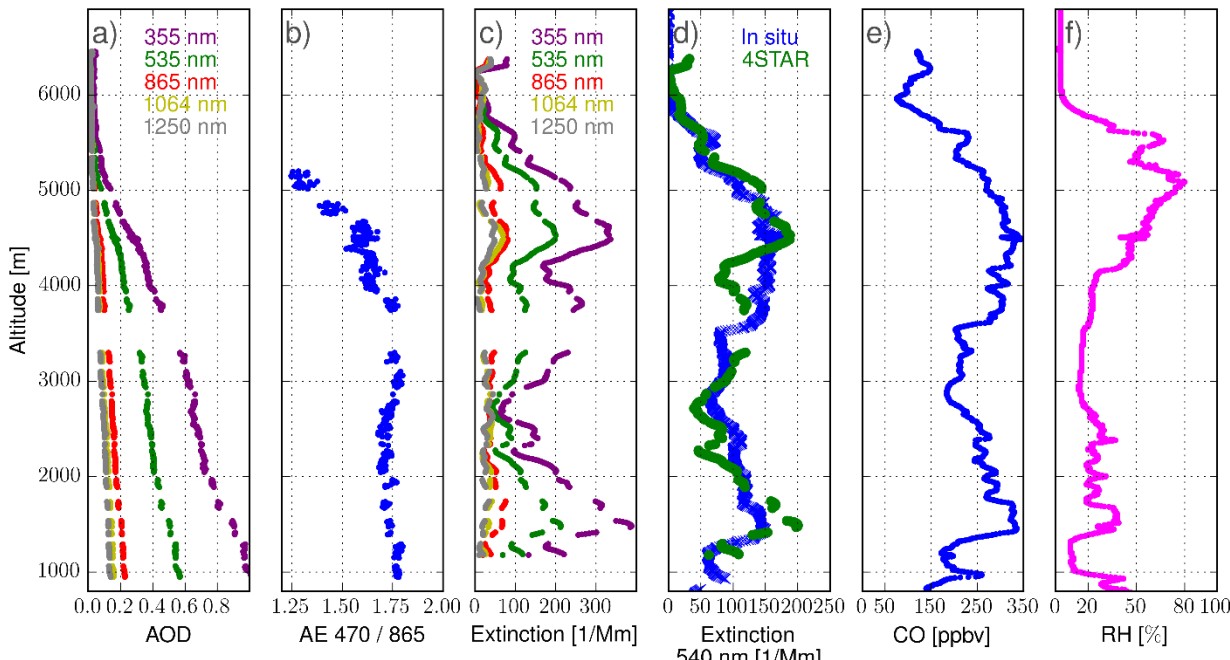

*Figure 13 – Aerosol optical properties profiles from the same case on 2016-09-20 as fig. 11. (a) Vertical profile of AOD at a few selected wavelengths. (b) $AE_{470/865}$ profile, (c) derived extinction coefficient from 4STAR AOD at a few wavelengths, (d) extinction coefficient at 540 nm derived from 4STAR AOD and in situ measurements, (e) CO concentration, (f) and ambient relative humidity (RH).*

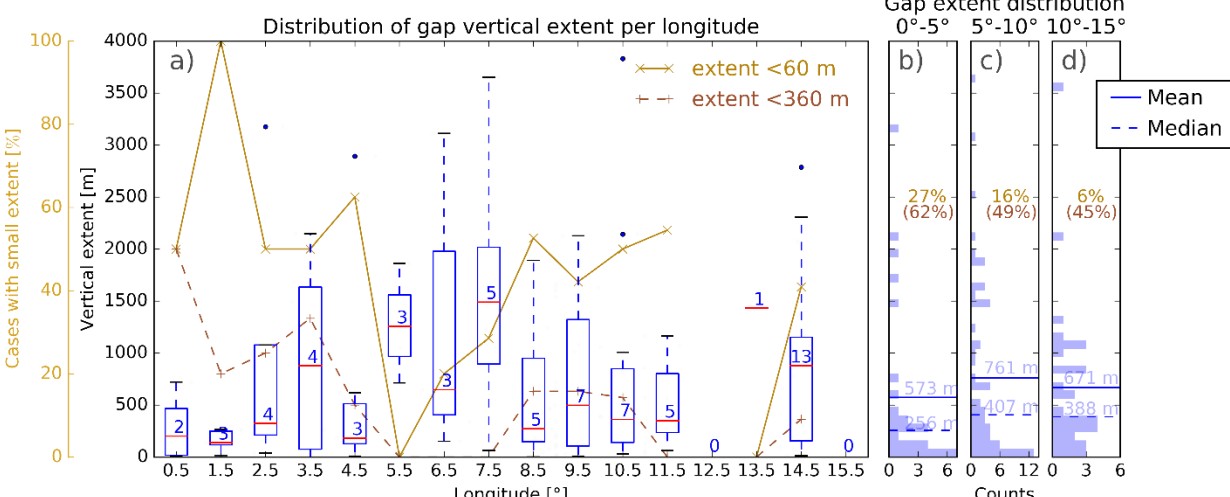

*Figure 14 - Distribution of vertical extent where the AOD does not change significantly with changing altitude (cloud-aerosol gap). (a) Box-whisker plot (red line representing mean of the bin, box representing the interquartile range, whiskers representing the minimum and maximum range, and outliers represented by dots, which are further than 1.5 times the interquartile range from the first or third quartile) of the vertical extent binned by longitude. Numbers indicate the number of days sampled represented within each bin, where each sampled day constitutes more than one profile. The proportion of sampled days that are considered having a small extent is denoted by the gold and brown colors. (b, c, and d) The gap altitude distribution represented as a histogram for all sampled ACAOD from 4STAR for 3 separate longitudinal regions. The proportion of the gap extent that is near zero is indicated as a percentage in each panel (b, c, and d), the equivalent statistic for CEAL cases (within 360 m) is below in parentheses.*

|  |  | All measurements | | | Spatially binned | | |
|---|---|---|---|---|---|---|---|
|  |  | mean | median | std | mean | median | std |
| ACAOD | 501 nm | 0.32 | 0.33 | 0.15 | 0.37 | 0.34 | 0.05 |
|  | 1020 nm | 0.09 | 0.09 | 0.05 | 0.11 | 0.09 | 0.02 |
| Total Column AOD | 501 nm | 0.36 | 0.30 | 0.18 | 0.38 | 0.39 | 0.03 |
|  | 1020 nm | 0.15 | 0.13 | 0.06 | 0.15 | 0.14 | 0.04 |
| ACAOD uncertainty | 501 nm | 0.011 | 0.01 | 0.008 | 0.013 | 0.011 | 0.004 |
|  | 1020 nm | 0.013 | 0.012 | 0.012 | 0.015 | 0.011 | 0.004 |
| AE of ACAOD | 470/865 nm | 1.71 | 1.75 | 0.24 | 1.65 | 1.66 | 0.10 |
|  | 500 nm | 1.45 | 1.48 | 0.18 | 1.44 | 1.48 | 0.06 |
| AE of Total Column | 470/865 nm | 1.25 | 1.30 | 0.46 | 1.23 | 1.33 | 0.09 |
|  | 500 nm | 1.08 | 1.14 | 0.37 | 1.07 | 1.19 | 0.07 |

*Table 1 - Summary of measured aerosol optical properties during September 2016 as part of ORACLES.*