# Peer review of "Above Cloud Aerosol Optical Depth from airborne observations in the South-East Atlantic"

_Atmospheric Chemistry and Physics, 2019_

## Short Comment (SC1) · 7 Feb 2019

A.M. Dzambo

dzamboam@gmail.com

This AMTD article highlights data collected during ORACLES 2016 using primarily the 4STAR instrument. The big takeaway I got from this was how much lower the ACAOD from 4STAR was compared to MODIS-based products, along with a general understanding of how ACAOD varies in the SE Atlantic using additional instrumentation. I will certainly be interested in learning more about why this is. The figures are beautiful, and the text is logically ordered and easy to follow. I have a few general questions and comments about the text that are hopefully easy to address & answer.

Page 6, Line 28: You should specify exactly what "high relative humidity" is here. Was it above 80%? 90%? It's certainly possible to have clouds in sub-saturated conditions,

and hence it would be useful to know the threshold you used in this backup in/out-of-cloud screening method.

Page 8, Line 28: I'm not entirely clear in understanding why the standard deviation of ACAOD is so large at longer wavelengths. Is it possible that there were fewer available quality measurements at those wavelengths? Also, why would signal-to-noise be lower?

Page 16, Line 5: Do you mean to say "the largest gap extents are observed near 7 degrees W"? I'm confused by what "the largest gap extents are observed not as expected but offshore" means. Perhaps it may be worth mentioning here where the largest gap extents are observed in the MOD06ACAOD vs. your measurements, to make this more clear in this section.

---

## Referee Comment (RC1) · Anonymous Referee #1 · 28 Feb 2019

The paper written by Samuel LeBlanc et al. presents the Above-Cloud AOT (ACAOD) measured by the 4STAR instrument during the September 2016 deployment of the OR-ACLES. The magnitude, the variability and the spatial distribution of the ACAOD and the Angstrom Exponent (AE) are analysed. The results are consistent with the location of the plume observed by satellites and the large AE values above clouds confirm the presence of small particles, linked to biomass burning. The spatial variation of the 4STAR ACAOD is compared with the clear-sky fine mode AOD and ACAOD retrieved from MODIS. Then, the vertical distributions of the measured ACAOD and AE are analysed, revealing the variability of the biomass burning plume position. For one vertical profile, the 4STAR measurements are compared with other instruments on-board the P3 aircraft, including CO concentration and extinction from in-situ measurements.

[Figure]

Finally, the distance between the aerosol layer and the cloud top is analysed and compared with previous satellite-based studies. The paper is clearly written and generally well-presented.

However, while the authors explain the spatial, vertical and spectral dependence of the measurements, I regret the absence of interpretation of the differences observed between the aircraft measurements and the satellite retrievals. As shown in the manuscript, the 4STAR ACAOD is found to be systematically lower than the satellite observations. Is the fine-mode AOD retrieved in clear-sky a good proxy for the ACAOD? Can the difference be explained by the contribution of the boundary layer to the MODIS fine mode AOD? Are the differences with MOD06ACAERO around 7W related to the variability of the AE observed by 4STAR? How does the AE of the MOD06ACAERO aerosol model compare with 4STAR? I also suspect the SSA of the MODIS aerosol above cloud model to have a key role in the ACAOD retrieval. Deaconu et al. (2017) have shown that aerosols within the clouds have an impact on the ACAOD retrieved from POLDER. Could it be the same with MODIS? Is there a correlation between the presence of an aerosol-cloud gap and the 4STAR/MODIS above-cloud differences? The aerosol-cloud gaps observed by 4STAR are compared to studies based on multi-year regional analysis of spaceborne lidars. Are the profiles sampled by the P3 representative of the South East Atlantic? ...

In my opinion, a discussion about the possible sources of discrepancies would be a valuable addition to this study, and I would recommend it for publication if this is addressed.

Specific remarks:

Page 2, lines 39-41: in the paper from Wen et al., the "cloud adjacency effect" describes the biases in the clear-sky AOD retrieved from passive satellites in-between clouds due to 3D contaminations. I am not sure how this applies to the aerosol above cloud retrievals. To my knowledge, the impact of 3D cloud effect on the ACAOD retrieval is

discussed in Peers et al. (2015) and Cornet et al. (2018). Also, the reference to Wen et al. (2007) is missing from the list.

Page 6, lines 1-7: which one of the two aerosol model assumptions is selected here? The MOD06ACAERO retrieval provides an estimation of the uncertainty. Also, the ACAOD retrieved by MODIS is expected to be less accurate for small cloud optical thicknesses (COT). Do the authors use any filters for the ACAOD based on the uncertainty or on the COT? This could have an impact of the results shown in figure 8.

Page 12, section 3.4.1: why did you use AOD profiles instead of extinction?

Page 12, section 3.4.2: it is worth stating in the text that the AE at a certain altitude represents the aerosol column located above this altitude. Is the AE calculated on the average AOD observed at that altitude, or is it an average of the observed AE? Does this include take-off and landing profiles in Walvis Bay? If this is the case, the AE profile from figure 10 tends to be overly representative of the aerosol variability in Walvis Bay. Also, I am surprised that the above-cloud AE profile stops at 4000m since the ACAOD profiles from figure 9 show the presence of aerosols until 6000m. Is it because the AOD is never larger than 0.1? For the "all data" profile, there are significantly more observations above 3000m than for the "above-cloud" profile, suggesting that most of the aerosol observations comes from clear-sky measurements. Unless the clear-sky profiles have been typically performed at different locations from the above-cloud profiles, I do not really understand why aerosols would be more frequently observed at higher altitudes in clear-sky than above clouds as I presume that the presence of aerosols at that altitude is not correlated to the absence or presence of clouds. Do you have any explanation?

Page 14, lines 17-19: it is also the location associated with the largest range. Could the maximum gap extents be due to a specific event, that may have been sampled on consecutive days?

Technical corrections:

Page 1, lines 40-41: "The peak ACAOD expected from long term retrievals . . ." I think this is a bit misleading, as one might think that this is the peak in the whole SEA region, while it is the peak observed along the routine flight path.

Page 2, line 20: ". . . the impact of these aerosolS . . ." Page 2, lines 21-22: ". . . from satellite measurements, where the ACAOD . . ." Page 2, lines 23-26: I understand that the paper is about the ORACLES measurements, but I would expect the authors to mention CLARIFY and AEROCLO-sA somewhere in the introduction.

Page 4, line 35: ". . . cloud DROPLET number concentration . . ."

Page 5, lines 6-8: apparently, the dry absorption is not corrected for humidity. What is the expected impact on the extinction?

Page 6, lines 37-39: do the measurements used here correspond to straight level run and/or vertical profiles between the cloud top and the aerosol layer? How is define a sample (see y-axis of fig. 3 and 6)? Does it correspond to measurements performed over a certain distance/time?

Page 7, lines 2-3: what is the maximum value of the ACAOD measured by 4STAR at 501 nm?

Page 7, lines 13-15: this sentence seems to imply that satellites would indifferently retrieved AOD in clear-sky and above clouds. Also, considering this formulation, it would be interesting to actually compare with the statistic obtained from MODIS.

Page 9, line 35: is the AE calculated from averaged AOD or is it averaged AE calculated on single measurement of the AOD?

Page 11, line 28: ". . . MOD06ACAERO . . ."

Page 12, lines 21-22: ". . . the far-from-coast versus profiles along the routine diagonal." Isn't it the same thing?

Page 13, lines 10-11: ". . . with minimal change in AOD being observed above that

altitude." Page 13, line 35: it might be useful to plot the relative humidity profile as well.

Page 16, line 22: ". . . throughout . . ."

References

Deaconu, L. T., et al. "Consistency of aerosols above clouds characterization from A-Train active and passive measurements." Atmospheric Measurement Techniques 10.9 (2017).

Peers, F., et al. "Absorption of aerosols above clouds from POLDER/PARASOL measurements and estimation of their direct radiative effect." Atmospheric Chemistry and Physics 15.8 (2015): 4179-4196.

Cornet, C., et al. "Cloud heterogeneity effects on cloud and aerosol above cloud properties retrieved from simulated total and polarized reflectances." Atmospheric Measurement Techniques 11 (2018).

Wen, G., et al. "3D aerosol cloud radiative interaction observed in collocated MODIS and ASTER images of cumulus cloud fields." Journal of Geophysical Research: Atmospheres 112.D13 (2007).

---

## Referee Comment (RC2) · Anonymous Referee #2 · 14 Mar 2019

The paper written by Samuel LeBlanc et al. is focused on the Aerosol Above Cloud (AAC) properties measured by the 4STAR instrument during the ORACLES 2016 airborne campaign. The magnitude and spatial variability, as well as the vertical distribution of above-cloud aerosol optical depth (ACAOD) and Ångström Exponent (AE) are discussed. A comparison of 4STAR ACAOD with satellite retrievals from MODIS of clear-sky fine mode AOD and ACAOD is also made. The authors also show a hyperspectral ACAOD profile case-study and a comparison of the 4STAR measurements with other in-situ instruments on-board of the P-3 aircraft. The final section is focused on the vertical distance between the aerosol layer and the underlying cloud. Their results show that during the ORACLES 2016 campaign the aerosol properties sampled above the clouds are consistent with previous studies, which show high elevated thick

layers of fine particles, typical to biomass burning aerosols. They also found that the largest ACAOD is found in the northern part of the sampling region, and that high AOD variability coincides with high AE variability. Both the satellite climatology over August and September and MODIS ACAOD retrievals collocated along the flight tracks consistently overestimate the 4STAR ACAOD. The final results show that the gap between the aerosol layers and the cloud tops is larger further away from the coast (0-4000 m) compared to near-coast samples (0-2500 m). The extent of the gap between aerosol and cloud peaks at a longitude of 7.5E, unlike the expected gradual decrease of the gap from the coast westwards.

Overall, the paper is well documented and generally well written, however there are several issues that should be further discussed in the paper. I would recommend it for publication if the following remarks are addressed.

General remarks:

While the authors have focused on the ACAOD retrievals with 4STAR, I was expecting a broader and more detailed analysis of aerosol properties, using other instruments onboard of the aircraft. For example, the aerosol absorption is not exploited even though the PSAP instrument has been used to compute the in-situ extinction. The in-situ aerosol extinction could have been compared for the entire sample set with the 4STAR derived extinction, since they seem to show some differences for one study case (Figure 12). It would had been interesting to see the difference in AOD from the 2 instruments. Also, the vertical aerosol profiles, and especially the distance between the aerosol layers and clouds could be further emphasized by showing a lidar profile, either from satellite retrievals or from the ER-2 aircraft. This could also give more confidence in the in-situ extinction measurements showed in Figure 2 and could support the "gap" definition for detached or attached situations.

I am not convinced of the statistical representability of the AOD full-column (and AE full column) from 4STAR, since there are so few samples, and if a comparison with the

ACAOD is bringing new insights. In most cases it is expected that a full-column AOD in this region should be higher than the ACAOD counterpart, due to the presence of sea salt aerosols. However, I am curious if the authors have considered the presence of POCs (pockets of open cells - Stevens et al., 2005; Wood et al., 2011) where the air is very clean and could had been possible that some of the full-column measurements were made within such a cell? That could perhaps add an explanation for the larger standard deviation observed.

In my opinion, there is a missing discussion regarding the variability in ACAOD around 8-10E and 18-22S and the larger variability of AE in the same region (page 10). Could it be possible that because the measurements were made at the southern region of the aerosol plume, the AOD during some measurements is small enough to impact the AE computation? We can clearly see how low AOT affects AE values in figure 12. Also, I am curious what could lead to such high standard deviation in the AOD retrievals (Figure 4), around 14S? It would be useful if the authors would add a color bar plot that would show the number of measurements averaged over each grid box of 0.6 x 0.65 deg, for both AOD and AE (if they are not the same).

While I appreciate the authors approach to compare with MODIS data, I found this section poorly explained. There are confusions related to the label of the MODIS products used for different comparisons; occasionally it is difficult to follow which comparison has been made. I suggest a rephrasing or a better description of these products. Also, it is not clear to me if the MODIS data were collocated along the flight track. What is the native resolution of the MODIS products? Have they been aggregated onto the grid box explained in section 3.1? From the description it appears that there is a subset of the MOD06ACAERO that has been temporally and spatially collocated with the flights, another subset that has been only spatially collocated (over the month of September), but the MODIS climatology dataset seems to be representative for the entire region. Is that right? I am also curious if the authors have investigated what happened on the date of 12/09, that shows anomalous values of the MODIS data? This is important,

since this high value seems to skew the mean AOD of that longitude bin in Fig. 8, and implicitly affect the analysis and conclusions of the paper.

Another concern is related to the analysis presented in section 3.5. It is not clear to me if this analysis was made along the routine flight track or using the entire dataset of measurements. If the samples are averaged along longitude bins from all the measurements, wouldn't that bias the results due to temporal and spatial variations of the vertical profile of the aerosol layer? The measurements were made on the extent of several days, in which the meteorological state or even diurnal cycle could have impacted the aerosols transport and their altitude. The authors should be more careful in making a statement on the expected aerosol layer transport and distance to cloud, particularly because of the meteorological uncertainty.

In my opinion, the paper could benefit from a little restructuration. Firstly, I consider that the study case can be presented as a methodology case, that leads further to the AE computation. The method of computing AE should also be presented in the methodology. I would also gather the AOD and AE sample statistics in one section, and the averaged spatial distribution of AOD and AE in another section. From my standpoint, this structure would make the paper easier to follow and clearer in its scientific message.

I suggest here a new plan, that the authors could consider.

1. Introduction

2. Data and instrumentation

2.1. ORACLES (Fig 1)

2.2. 4STAR

2.3. In-situ instruments

2.4. AERONET

2.5. Satellite retrievals

3. Methodology

3.1. AOD above cloud determination (Fig 13 + Fig 2)

3.2. Spectral AOD above cloud and AE

3.2.1. Hyperspectral AOD study-case (Fig 11 + Fig 12)

3.2.2. AE estimation (Fig 5)

4. Results

4.1. Statistics of sampled AOD and AE (Fig 3 + Fig 6)

4.2. Spatial distribution of AOD and AE (Fig 4 + Fig 7 + figure with number of measurements / grid box)

4.3. Airborne AOD in context of climatology and satellite measurements (Fig 8)

4.4. Aerosol vertical profiles and distance to clouds

4.4.1. Spatial variability in AOD profiles (Fig 9)

4.4.2. AE vertical dependence (Fig 10)

4.4.3. AOD distance to cloud (Fig 14)

5. Summary and discussion

Specific remarks:

Page 5, lines 5-10: The absorption coefficient is taken in dry conditions, while the scattering coefficient is measured at different relative humidity conditions. How does that impact the computation of the extinction coefficient and what is the resulted uncertainty?

Line 43: What is the minimum cloud optical thickness for which the MODIS product

can retrieve AOD above cloud? Do you know what are the uncertainties of the MODIS ACAOD product for optically thin clouds (e.g. COT < 5)?

Page 6, line 5: "two aerosol model assumptions..." What are those assumptions? Lines 16-20: Maybe move Page 14, lines 8-11 here for better clarification. Do you know how the measurement uncertainty (60 m) for "no gap" situations affect the ACAOD retrieval?

Page 7, lines 13-15: "Considered together, the ACAOD and full column AOD (denoted by the total extent of the histogram bars in Fig. 3) represent what a satellite remote sensor would retrieve in the region, if it were spatially and temporally co-located to the NASA P-3 aircraft." I understand this sentence, but I don't understand its relevance to the paper.

Page 9, line 3: Why are you fitting the AE up to 1650 nm, since you have showed that for wavelengths larger than 1000 nm, the AOD has higher uncertainties (Fig. 5)?

Lines 25 – 27: You mention the negative difference of above cloud AE500 -AE470/865 of -0.26 as representative to biomass burning sources, as defined by Yoon et al., 2012. I do not understand why the full column difference is positive? Aren't you subtracting AE470/865 (=1.25) from AE500 (=1.08) which would result in -0.17? I am confused of your conclusion for this analysis. Could you please clarify it in the text?

Page 10, line 33: Why are you using August in the MODIS AOD climatology? As shown is Adebiyi et al., (2015) there is a shift in meteorological condition between August and September, and this would result in a change in aerosol transport – hence climatology.

Page 12, Section 3.4.1: - Figure 9 c): how come there are profiles that begin almost at 0 m? Shouldn't these be measurements above the cloud? Are these all column AODs or ACAODs?

- "[...] the variability of the AOD profile in these different regions, we observed at 2000 m AOD ranges between 0.17 to 0.6 for 25 profiles along the routine diagonal, and 0.3 to

0.58 for coastal profiles." Are you discussing only figures 9c) and 9d)? If you consider all the plots, then at 2000 m I observe larger values of AOD closer to the coast.

Page 12, Section 3.4.2.: I do not understand how there are more data at 3000-4000 m for "all data" situations in Figure 10 compared to "above cloud" situations. Wouldn't the measurements at that altitude include only above-cloud measurements? Why is the AE above cloud profile stopping at 4000 m, when the ACAOD goes up to 6000 m (Fig. 9)? All the AOD above 4000 m is < 0.1, and thus you have filtered these data?

Figure 1: Add the latitude and longitude coordinates

Figure 14 is confusing. You talk about profiles in the text, but show days on the plot. You should mention one or the other in the text or on the plot.

Technical corrections:

General observation: I observed missing commas in many places, which makes the text difficult to follow in some cases.

Page 2, line 41: Wen et al., (2017) reference is missing. I also suggest Cornet et al., (2018) that looked at POLDER 3-D cloud radiative effects.

Page 3, Line 1: You could also mention Hu et al., (2007) for the CALIOP Depolarization Ratio Method, since you mention Chand et al., (2009) for CALIOP Color-Ratio Method.

Line 1: Matus et al. 2015 reference is missing

Lines 11-15: Maybe split this sentence, as it is hard to follow: "Past work has shown that the elevated aerosol layers in this region are frequently separated from the underlying cloud top, e.g., Devasthale and Thomas (2011) found that 90-95

Page 5, line 1-3: Rephrase: "with paired single-wavelength nephelometers (Radiance Research M903 measuring at 540 nm, with air in one humidified to 80

Line 31: You have already mentioned Gobabeb and Henties Bay in line 26. Maybe

rephrase a bit so you don't repeat the same information.

Page 6, Line 3: "...uses reflectance ..."

Lines 40-42: Change "[. . .] including the biomass burning layer of aerosol above clouds as well as any lower-level aerosol near the sea surface." to "including the elevated biomass burning layer as well as any lower-level aerosol near the sea surface."

Page 9, Line 11: Kaufman 1993, missing reference

Line 42: "[. . .] in the southern part of the sampling" should be correct

Page 11: lines 11-13: "The peak August and September mean ACAOD from MOD06ACAERO at the most western edge of the region, near 0°E, is shifted to the east in the subsampled MOD06ACAERO." – unclear of which product you are referring to

Lines 15-18: this sentence is too long "The largest [. . .] monthly statistics"

Page 12, line 20-22: "[. . .] large variability is noticeable, especially when contrasting the far-from-coast versus profiles along the routine diagonal". High variability of? Also, I thought far-from-coast is along the routine diagonal, thus this sentence is not clear.

Page 14, line 5: there is no Section 3.2.3

Line 12: Sakaeda et al., 2011: This study does not analyses the aerosol-cloud distance (longitudinal gap) from spaceborne lidar. Could you replace your reference with the right one?

Page 15, line 15: add reference for the MODIS AOD 12 years climatology

Line 23: "For the full column AOD, the AE470/865 is much lower than its above cloud counterpart". It is not much lower, since the value of the mean difference between ACAOD and AOD equals 0.4, which is within the uncertainty range (standard deviation of ACAOD is up to 0.4 – Figure 7).

Line 32: You could add a discussion of the following sentence: "The ACAOD from 4STAR also has a peak closer to shore than the MODIS AOD climatology mean and median (both fine and coarse mode), with differences near coast between 4STAR ACAOD measurements and MOD06ACAERO retrievals."

Page 16, lines 10-12: "From these airborne measurements, we have seen that the ACAOD is lower than expected from current MODIS satellite retrievals during the measurement period (by 0.05-0.08) and from a 12-year climatology (by 0.04)." Unclear sentence. Do you mean ". . .compared to current MODIS satellite retrievals. . ."?

References:

Adebiyi, A. A., Zuidema, P. and Abel, S. J.: The Convolution of Dynamics and Moisture with the Presence of Shortwave Absorbing Aerosols over the Southeast Atlantic, J. Clim., 28(5), 1997–2024, doi:10.1175/JCLI-D-14-00352.1, 2015.

Cornet, C., C.-Labonnote, L., Waquet, F., Szczap, F., Deaconu, L., Parol, F., Vanbauce, C., Thieuleux, F., and Riédi, J.: Cloud heterogeneity on cloud and aerosol above cloud properties retrieved from simulated total and polarized reflectances, Atmos. Meas. Tech., 11, 3627-3643, https://doi.org/10.5194/amt-11-3627-2018, 2018.

Stevens, B., G. Vali, K. Comstock, R. Wood, M.C. van Zanten, P.H. Austin, C.S. Bretherton, and D.H. Lenschow, 2005: POCKETS OF OPEN CELLS AND DRIZZLE IN MARINE STRATOCUMULUS. Bull. Amer. Meteor. Soc., 86, 51–58, https://doi.org/10.1175/BAMS-86-1-51

Wood, R., Bretherton, C. S., Leon, D., Clarke, A. D., Zuidema, P., Allen, G., and Coe, H.: An aircraft case study of the spatial transition from closed to open mesoscale cellular convection over the Southeast Pacific, Atmos. Chem. Phys., 11, 2341-2370, https://doi.org/10.5194/acp-11-2341-2011, 2011.

---

## Referee Comment (RC3) · Anonymous Referee #3 · 9 Apr 2019

The paper "Above Cloud Aerosol Optical Depth from airborne observations in the South-East Atlantic" by LeBlanc et al., presents observations of aerosol above the cloud in a region characterized by the presence of biomass burning emitted aerosol. Many interesting results are reported in the paper, but in my opinion the presentation is confusing in some points and this makes the paper a difficult reading. Some points can be substantially improved in clearness allowing to reach more large audience.

Here some suggestions about things to be improved for making it a very good paper:

1) in the introduction authors correctly underlined the importance of the vertical clean air between aerosol and clouds and how this is important for radiation budget issues: because of this one would expect an analysis of the results in this respect . I would suggest authors to include this into the discussion otherwise (not suggested) please

remove this from the introduction

2) in the introduction is stated that active technique can provide a very good insight about the aerosol/cloud gaps, but then authors used ORACLES dataset without explaining why and which are the added value in doing that. Reading the introduction one has the impression that the in depth analysis is elsewhere reported e.g. in the lidar papers.

3) assumption about fine mode as representative of the ACAOD has to be discussed. This could aslo lead to the differences observed between the ACAOD here presented and the MODIS data

4) discussion about figures 6: these indicates also that the Angstrom exponent changes a lot when the total column is considered even if the difference in AOD is not so relevant. Please comment on that and provide explanation of this aspect

5) fig 9 (and 12) these profiles of AOD would like to simulate the AOD as observed from space? It seems to me the integration of extinction is made from the above to the ground. Typically profiles are reported for extinction and not for AOD which is columnar quantity and not range resolved. This is misleading for the reader.

6) not clear why there is a big difference in AE above 2 km for above the cloud and total column cases (fig10). Please analyze and explain this

More detailed comments are reported as comments into the attached pdf

Please also note the supplement to this comment:
https://www.atmos-chem-phys-discuss.net/acp-2019-43/acp-2019-43-RC3-supplement.pdf

---

## Author Comment (AC1) · 21 Aug 2019

We appreciate comments from the reviewers and the push to enhance this manuscript's quality. Please see below the responses to each reviewer in blue italic. We have added discussions spanning most of the reviewers' comments, with emphasis on descriptions of the satellite products and their discussion in the frame of our measurements presented here. We have also adjusted multiple figures with the reviewers' comments in mind, and the overall structure of the paper has been refined. In line with the spirit of the reviewer's comments, we have enhanced the writing quality throughout the manuscript. See below for more details.

**Anonymous Referee # 1**

[Figure]

The paper written by Samuel LeBlanc et al. presents the Above-Cloud AOT (ACAOD) measured by the 4STAR instrument during the September 2016 deployment of the OR-ACLES. The magnitude, the variability and the spatial distribution of the ACAOD and the Angstrom Exponent (AE) are analysed. The results are consistent with the location of the plume observed by satellites and the large AE values above clouds confirm the presence of small particles, linked to biomass burning. The spatial variation of the 4STAR ACAOD is compared with the clear-sky fine mode AOD and ACAOD retrieved from MODIS. Then, the vertical distributions of the measured ACAOD and AE are analysed, revealing the variability of the biomass burning plume position. For one vertical profile, the 4STAR measurements are compared with other instruments on-board the P3 aircraft, including CO concentration and extinction from in-situ measurements.

Finally, the distance between the aerosol layer and the cloud top is analysed and compared with previous satellite-based studies. The paper is clearly written and generally well-presented.

However, while the authors explain the spatial, vertical and spectral dependence of the measurements, I regret the absence of interpretation of the differences observed between the aircraft measurements and the satellite retrievals. As shown in the manuscript, the 4STAR ACAOD is found to be systematically lower than the satellite observations. Is the fine-mode AOD retrieved in clear-sky a good proxy for the ACAOD? Can the difference be explained by the contribution of the boundary layer to the MODIS fine mode AOD? Are the differences with MOD06ACAERO around 7W related to the variability of the AE observed by 4STAR? How does the AE of the MOD06ACAERO aerosol model compare with 4STAR? I also suspect the SSA of the MODIS aerosol above cloud model to have a key role in the ACAOD retrieval. Deaconu et al. (2017) have shown that aerosols within the clouds have an impact on the ACAOD retrieved from POLDER. Could it be the same with MODIS? Is there a correlation between the presence of an aerosol-cloud gap and the 4STAR/MODIS above-cloud differences?

The aerosol-cloud gaps observed by 4STAR are compared to studies based on multi-year regional analysis of spaceborne lidars. Are the profiles sampled by the P3 representative of the South East Atlantic?

*For the questions:* "*Is the fine-mode AOD retrieved in clear-sky a good proxy for the ACAOD? Can the difference be explained by the contribution of the boundary layer to the MODIS fine mode AOD?*" *– We have included in the discussion of section 3.3 (now 4.3) in the before last paragraph:* "*The assumption that all fine mode AOD in clear sky retrieved by MODIS over 12 years is representative of the above cloud AOD should be revisited. This assumes that 1) no aerosol in the marine boundary layer contributes to the fine mode AOD and 2) aerosol in clear sky is representative of the above cloud aerosol. As far as the first assumption is concerned, a polluted marine boundary layer with non-negligible black carbon concentrations was observed during at times ORACLES 2016 (ORACLES Science Team, 2017), which would indicate that the proxy ACAOD from MODIS 12 year climatology may be an upper bound of the ACAOD.*"

*For the question:* "*Are the differences with MOD06ACAERO around 7W related to the variability of the AE observed by 4STAR?*" *– We added a paragraph (the fifth in the revised document) in section 4 (now 5) to discuss this and other questions, with the explanation* "*The regions where the largest divergence between MOD06ACAERO coincide with the largest variability in AE (near 7° E), likely indicates a link between aerosol properties and the accuracy of MOD06ACAERO. Complicating factors in this region may be linked to the occurrence of mid-level clouds topping the aerosol layer, which have been observed in this region and has also been reported, in the form of elevated RH, to occur over a longer time sample from satellite and sounding observations by Adebiyi et al., (2015).*"

*For the question and comment: "How does the AE of the MOD06ACAERO aerosol model compare with 4STAR? I also suspect the SSA of the MODIS aerosol above cloud model to have a key role in the ACAOD retrieval." – In the newly added paragraph, we summarize the comparison to the aerosol model AE: "Here we found a smaller AE470/865 (mean: 1.71), than what is defined in the aerosol model within the MO06ACAERO retrieval ($\sim$ 2.0 when the AOD at 550 nm is 0.5 from Levy et al., 2007 , with an AOD dependence), which may suggest the underlying aerosol model needs refinement."*

*For the questions: "Deaconu et al. (2017) have shown that aerosols within the clouds have an impact on the ACAOD retrieved from POLDER. Could it be the same with MODIS? Is there a correlation between the presence of an aerosol-cloud gap and the 4STAR/MODIS above-cloud differences?" – We have included this discussion: "Differences between MOD06ACAERO and 4STAR ACAOD may also be attributable to satellite retrieval sensitivities to aerosol embedded within clouds, although these differences do not seem to correlate with the gap extent. Embedded aerosol within clouds is still possible through the inclusion of marine boundary layer aerosols mixing upwards in clouds, or that above cloud aerosol have mixed into underlying clouds, but at a past period in the cloud's lifetime (Diamond et al., 2018). Other possible sources of differences may be the underlying selection of aerosol model (aerosol single scattering albedo, asymmetry parameter, etc.) in the MODIS ACAOD retrieval or the cloud mask applied (i.e., only using cloud of optical thickness 4 and above)."*

*For the last comment and question: "The aerosol-cloud gaps observed by 4STAR are compared to studies based on multiyear regional analysis of spaceborne lidars. Are the profiles sampled by the P3 representative of the South East Atlantic?" – We believed that this analysis is outside the*

*scope of this paper, which would require a high temporal comparison of li-
dar and/or model data. We summarize our comments in a newly added last
paragraph of section 3.5 (now 4.5):"* The exact representativeness of these
results, including the aerosol layer vertical distribution, from airborne sam-
pling to the natural world are investigated in future studies (e.g., Shinozuka
et al., Submitted to ACP). There is likely a large inter-annual variability and
geographical sampling variations in the SEA, which could skew the com-
parison between airborne and satellite sampling."

In my opinion, a discussion about the possible sources of discrepancies would be
a valuable addition to this study, and I would recommend it for publication if this is
addressed.

Specific remarks:

Page 2, lines 39-41: in the paper from Wen et al., the "cloud adjacency effect" de-
scribes the biases in the clear-sky AOD retrieved from passive satellites in-between
clouds due to 3D contaminations. I am not sure how this applies to the aerosol above
cloud retrievals. To my knowledge, the impact of 3D cloud effect on the ACAOD re-
trieval is discussed in Peers et al. (2015) and Cornet et al. (2018). Also, the reference
to Wen et al. (2007) is missing from the list.

*Restructured this section slightly, while adding references to Peers et al
and Cornet et al. Now reads:* "However, current passive satellite ACAOD
retrieval techniques could be biased by what is called the "*cloud adjacency
effect*" (Wen et al., 2007) or the "*3-D cloud radiative effect*", i.e., brighten-
ing of cloud-free air near clouds, that also extends to above cloud aerosol
properties, which has been observed using polarized light (Cornet et al.,
2018). 3-D cloud radiative effects also impact retrievals of aerosol above

*clouds, where the underlying cloud heterogeneity impact the aerosol sub-jected radiance (Peers et al., 2015). Wen et al., 2007 reference added.*

Page 6, lines 1-7: which one of the two aerosol model assumptions is selected here?

The MOD06ACAERO retrieval provides an estimation of the uncertainty. Also, the ACAOD retrieved by MODIS is expected to be less accurate for small cloud optical thicknesses (COT). Do the authors use any filters for the ACAOD based on the uncertainty or on the COT? This could have an impact of the results shown in figure 8.

*We specified which aerosol model is used here (MOD04; Levy et al., 2009). And we added this sentence to the description to better instruct the reader on the filters used: "Consistent with Meyer et al. (2015), we report only the AOD from MOD06ACAERO above clouds with an optical thickness of greater than 4, and AOD uncertainties lower than 100%." . This filtering may in fact cause some impact to results in figure 8 (now fig. 9). From this comment we added this sentence to the discussion at the end of new sect. 4.3 : "Additionally, the filtering of MOD06ACAERO, to only apply retrievals over opaque water clouds (with optical thicknesses greater than 4), may lead to systemic biases in ACAOD."*

Page 12, section 3.4.1: why did you use AOD profiles instead of extinction?

*Since the derivation of extinction profiles have compounding uncertainty as compared to the primary measurement of AOD, the authors did not see the added interpretation benefit of using slightly higher uncertain profiles of extinction coefficients.*

Page 12, section 3.4.2: it is worth stating in the text that the AE at a certain altitude represents the aerosol column located above this altitude. Is the AE calculated on the average AOD observed at that altitude, or is it an average of the observed AE? Does this include take-off and landing profiles in Walvis Bay? If this is the case, the AE profile from figure 10 tends to be overly representative of the aerosol variability in Walvis Bay. Also, I am surprised that the above-cloud AE profile stops at 4000m since the ACAOD profiles from figure 9 show the presence of aerosols until 6000m. Is it because the AOD is never larger than 0.1? For the "all data" profile, there are significantly more observations above 3000m than for the "above-cloud" profile, suggesting that most of the aerosol observations comes from clear-sky measurements. Unless the clearsky profiles have been typically performed at different locations from the above-cloud profiles, I do not really understand why aerosols would be more frequently observed at higher altitudes in clear-sky than above clouds as I presume that the presence of aerosols at that altitude is not correlated to the absence or presence of clouds. Do you have any explanation?

*We have added clarification to section 3.4.2 (now 4.4.2) to address these questions. The authors hope that it is now clearer that all data include partial aerosol columns, while ACAOD only includes entire above cloud aerosol layers. The new section reads: "Considering all measurements made during ORACLES 2016 from the P-3, the AE470/865*

*is roughly constant at a median value of 1.75 for the column of aerosol extending from base altitudes ranging between 600 m and 6 km to the top of atmosphere, whereas for column bases below that, the median decreases monotonically to 0.6 (Figure 11).The AE flagged as ACAOD (blue colors, fig. 11) is calculated from individual AOD spectra only for the portions encompassing the entirety of the above cloud aerosol layer. The AE for all data is calculated from AOD spectra representing aerosol above the aircraft altitude, often only partially representing aerosol layers, regardless if there are clouds or aerosol in the underlying column. The inclusion of all data permits the quantification of AE at altitudes higher than the highest base altitude of aerosol above cloud layer(s) (which is just shy of 4000 m)." For the representation of Walvis Bay, we included a note in the manuscript: ". . . particles near sea surface, and is reproduced over more than 9 days sampled, even when filtering out the profiles near Walvis Bay (not shown), where there was significant dust." Hopefully this description showcases the differences in interpretation of AE at different altitudes, such that it is expected that all data has AE up to 6000 m, but not for ACAOD.*

Page 14, lines 17-19: it is also the location associated with the largest range. Could the maximum gap extents be due to a specific event, that may have been sampled on consecutive days?

*This has been changed to make it more specific: "[. . . ] and is observed over 5 non-consecutive days spanning 8/31 to 9/20, with gaps larger than 1km observed on 9/06 at 18.2° S, on 9/10 at 17.8° S, and on 9/14 at 16.1° S to 17.7° S."*

Technical corrections:

Page 1, lines 40-41: "The peak ACAOD expected from long term retrievals . . . " I think this is a bit misleading, as one might think that this is the peak in the whole SEA region, while it is the peak observed along the routine flight path.

*Added the caveat: "AOD along a diagonal routine track extending out from the coast of Namibia" within the sentence.*

Page 2, line 20: ". . . the impact of these aerosolS . . . " Page 2, lines 21-22: ". . . from satellite measurements, where the ACAOD . . . " Page 2, lines 23-26: I understand that the paper is about the ORACLES measurements, but I would expect the authors to mention CLARIFY and AEROCLO-sA somewhere in the introduction.

*Adjusted the grammar mistakes and included a sentence near the end of the introduction: "in conjunction with other large scale field missions focused in the same region; CLARIFY [CLoud – Aerosol – Radiation InteRactions and Forcing for Year 2017;Zuidema et al., 2016], AEROCLO-sA [AErosols, RadiatiOn and CLOuds in southern Africa; Formenti et al., 2019], and LASIC [Layered Atlantic Smoke Interactions with Clouds; Zuidema et al., 2018]."*

Page 4, line 35: ". . . cloud DROPLET number concentration . . . "

*Corrected.*

Page 5, lines 6-8: apparently, the dry absorption is not corrected for humidity. What is the expected impact on the extinction?

*This question continues to be an open discussion in the scientific literature
and is well outside the scope of this paper.*

Page 6, lines 37-39: do the measurements used here correspond to straight level run
and/or vertical profiles between the cloud top and the aerosol layer? How is define a
sample (see y-axis of fig. 3 and 6)? Does it correspond to measurements performed
over a certain distance/time?

*Modified sentence (now start of sect. 4.1) to: "We have separated all
4STAR measurements in the SEA into either ACAOD (11.5 hours of mea-
surements, from flags described in section 2.1) or full column AOD (0.9
hours of measurements in level legs or profiles below 600 m in altitude)."
And added: "Figure 4 shows the distribution of those measurements, with
roughly 1 sample per second, at two wavelengths."*

Page 7, lines 2-3: what is the maximum value of the ACAOD measured by 4STAR at
501 nm?

*Added "with an absolute range of 0.02 to 1.04" .*

Page 7, lines 13-15: this sentence seems to imply that satellites would indifferently
retrieved AOD in clear-sky and above clouds. Also, considering this formulation, it
would be interesting to actually compare with the statistic obtained from MODIS.

*Added this caveat to the sentence: "if satellite retrievals would not discrimi-
nate between full column and over clouds" – Further comparison is outside*

*the scope of this paper, and is meant to invite future analysis based on this result.*

Page 9, line 35: is the AE calculated from averaged AOD or is it averaged AE calculated on single measurement of the AOD?

*Added in last paragraph of section 4.2: "calculated from each AOD measurement" .*

Page 11, line 28: ". . . MOD06ACAERO . . . "

*Corrected.*

Page 12, lines 21-22: ". . . the far-from-coast versus profiles along the routine diagonal." Isn't it the same thing?

*Changed to "near-coast" instead of "far-from-coast" .*

Page 13, lines 10-11: ". . . with minimal change in AOD being observed above that altitude."

*Corrected.*

Page 13, line 35: it might be useful to plot the relative humidity profile as well.

*A panel showing the relative humidity has been added to Figure 12 (now fig. 13), with the associated text changes: "The relative humidity for this profile is between 10% and 80% within the aerosol layers (Fig. 13f), with the majority of the profile near 20% RH ".*

Page 16, line 22: "... throughout ... "

*Unchanged, we do mean throughput; the change of light passing through the spectrometers.*

References

Deaconu, L. T., et al. "Consistency of aerosols above clouds characterization from ATrain active and passive measurements." Atmospheric Measurement Techniques 10.9 (2017).

Peers, F., et al. "Absorption of aerosols above clouds from POLDER/PARASOL measurements and estimation of their direct radiative effect." Atmospheric Chemistry and Physics 15.8 (2015): 4179-4196.

Cornet, C., et al. "Cloud heterogeneity effects on cloud and aerosol above cloud properties retrieved from simulated total and polarized reflectances." Atmospheric Measurement Techniques 11 (2018).

Wen, G., et al. "3D aerosol cloud radiative interaction observed in collocated MODIS and ASTER images of cumulus cloud fields." Journal of Geophysical Research: Atmospheres 112.D13 (2007).

**Anonymous Referee # 2**

The paper written by Samuel LeBlanc et al. is focused on the Aerosol Above Cloud (AAC) properties measured by the 4STAR instrument during the ORACLES 2016 airborne campaign. The magnitude and spatial variability, as well as the vertical distribution of above-cloud aerosol optical depth (ACAOD) and Ångström Exponent (AE) are discussed. A comparison of 4STAR ACAOD with satellite retrievals from MODIS of clear-sky fine mode AOD and ACAOD is also made. The authors also show a hyperspectral ACAOD profile case-study and a comparison of the 4STAR measurements with other in-situ instruments on-board of the P-3 aircraft. The final section is focused on the vertical distance between the aerosol layer and the underlying cloud. Their results show that during the ORACLES 2016 campaign the aerosol properties sampled above the clouds are consistent with previous studies, which show high elevated thick layers of fine particles, typical to biomass burning aerosols. They also found that the largest ACAOD is found in the northern part of the sampling region, and that high AOD variability coincides with high AE variability. Both the satellite climatology over August and September and MODIS ACAOD retrievals collocated along the flight tracks consistently overestimate the 4STAR ACAOD. The final results show that the gap between the aerosol layers and the cloud tops is larger further away from the coast (0-4000 m) compared to near-coast samples (0-2500 m). The extent of the gap between aerosol and cloud peaks at a longitude of 7.5E, unlike the expected gradual decrease of the gap from the coast westwards.

Overall, the paper is well documented and generally well written, however there are several issues that should be further discussed in the paper. I would recommend it for publication if the following remarks are addressed.

General remarks:

While the authors have focused on the ACAOD retrievals with 4STAR, I was expecting

a broader and more detailed analysis of aerosol properties, using other instruments onboard of the aircraft. For example, the aerosol absorption is not exploited even though the PSAP instrument has been used to compute the in-situ extinction. The in-situ aerosol extinction could have been compared for the entire sample set with the 4STAR derived extinction, since they seem to show some differences for one study case (Figure 12).

*We appreciate the sentiment of including much more aerosol intensive property comparisons, but this is outside the scope of this paper. We would like to focus the reviewer to the newly published manuscript by Pistone et al. (2019) which goes in depth with understanding a single aerosol intensive property (single scattering albedo) as viewed by different sensors. To accommodate the comparison of the case study, we have included a profile of relative humidity, to help elucidate the differences between AOD and the in situ extinction profile.*

*Pistone, K., Redemann, J., Doherty, S., Zuidema, P., Burton, S., Cairns, B., Cochrane, S., Ferrare, R., Flynn, C., Freitag, S., Howell, S. G. and Kacenelenbogen, M.: Intercomparison of biomass burning aerosol optical properties from in situ and remote-sensing instruments in ORACLES-2016, Atmos. Chem. Phys., 19, 9181–9208, doi:10.5194/acp-19-9181-2019, 2019.*

It would had been interesting to see the difference in AOD from the 2 instruments. Also, the vertical aerosol profiles, and especially the distance between the aerosol layers and clouds could be further emphasized by showing a lidar profile, either from satellite retrievals or from the ER-2 aircraft. This could also give more confidence in the in-situ extinction measurements showed in Figure 2 and could support the "gap" definition for detached or attached situations.
*Extinction profiles are not directly measured by either PSAP or 4STAR, which would lead to an analysis of the errors associated with extinction profile calculations, diverging from the focus of this paper which is the Above Cloud AOD. The interesting work of an AOD closure study with in situ and remote sensing is being pursued separately to give it the full attention it deserves. Similarly, comparisons to Lidar profiles may lead to investigations in the differences, and steer away from the focus of the paper, although initial case study comparisons are quite promising. For comparing to lidar, there is the in depth question of colocation with lidar measurements either from the ER-2 aircraft, or from satellites (CALIPSO). Figure 2 (now fig. 3) uses combination of scattering coefficient and AOD profiles but not extinction for defining the gap extent.*

I am not convinced of the statistical representability of the AOD full-column (and AE full column) from 4STAR, since there are so few samples, and if a comparison with the ACAOD is bringing new insights. In most cases it is expected that a full-column AOD in this region should be higher than the ACAOD counterpart, due to the presence of sea salt aerosols. However, I am curious if the authors have considered the presence of POCs (pockets of open cells - Stevens et al., 2005; Wood et al., 2011) where the air is very clean and could had been possible that some of the full-column measurements were made within such a cell? That could perhaps add an explanation for the larger standard deviation observed.

*We have introduced a new panel in Figure 4 (now fig. 5), and a paragraph to help address this issue. See Figure 5c and the related paragraph in section 4.1: "The full column AOD501 sampled by 4STAR and AERONET locations is presented in Fig. 5c, where its paucity of samples is apparent, particularly in the middle of sampling region where ACAOD shows higher*

*than average values. The occasions where the P3 sampled full column AOD occurred nearly always at the edges of the cloud layers These full column measurements were not inside pockets of open cells clouds (POC; Stevens et al., 2005; Wood et al., 2011). Full column AOD measurements were more commonly measured past the southern edge of the stratocumulus cloud deck, and where the marine boundary layer was both polluted by biomass burning or with a clean background (ORACLES Science Team, 2017). Where a direct comparison of the full column AOD and the ACAOD is possible, the full column AOD501 is higher by an average of 0.03 (mean full column AOD501 is 0.38 vs. mean ACAOD501 is 0.35 at the same locations).This difference is nearly reproduced by AERONET, impacted by dust and sea salt in the boundary layer over land with overlying biomass burning aerosol, in the average fine mode AOD501 (0.2) and total AOD501 (0.24).*"

In my opinion, there is a missing discussion regarding the variability in ACAOD around 8-10E and 18-22S and the larger variability of AE in the same region (page 10). Could it be possible that because the measurements were made at the southern region of the aerosol plume, the AOD during some measurements is small enough to impact the AE computation? We can clearly see how low AOT affects AE values in figure 12. Also, I am curious what could lead to such high standard deviation in the AOD retrievals (Figure 4), around 14S? It would be useful if the authors would add a color bar plot that would show the number of measurements averaged over each grid box of 0.6 x 0.65 deg, for both AOD and AE (if they are not the same).

*We looked into more detail for the aerosol around 8-10E an 18-22S, and we have concluded new observations, summarize at the end of that section, which now reads: "The high standard deviation in AE in this region*

*is associated with ACAOD between 0.2 and 0.45 with AE from 0.2 to 1.2. These aerosols, sampled over more than one day, may not be uniquely biomass burning, but the low AE may indicate that there is water vapor condensation on aerosol by neighboring mid-level clouds, observed in few flights in that region." Although the AE values in figure 12 (now fig. 13), it may not uniquely due to the low bias of AE at lower AOD. We cut off the AE reporting at low AOD, to mitigate displaying the low AE bias. In other cases we have looked at, the decreasing trend of AE with AOD, with this limit, is not always present.*

*The size of the boxes denote the number of days sampled within that box, they are the same for both ACAOD and AE, we clarified this in the caption of Figure 7 (now fig. 8): "[. . . ] represents the number of sampling days used to build the statistics within each gridded bins, nearly the same number of samples as shown in Fig. 5a." We have gone back and forth between only putting the number of samples in each box, but it seemed less informative, as the sampling rate is 1Hz, but geographical representation is spurious. We have updated the figure (fig. 5a) to also show the number of samples within each bin, represented as circles with varying sizes.*

While I appreciate the authors approach to compare with MODIS data, I found this section poorly explained. There are confusions related to the label of the MODIS products used for different comparisons; occasionally it is difficult to follow which comparison has been made. I suggest a rephrasing or a better description of these products. Also, it is not clear to me if the MODIS data were collocated along the flight track. What is the native resolution of the MODIS products? Have they been aggregated onto the grid box explained in section 3.1? From the description it appears that there is a subset of the MOD06ACAERO that has been temporally and spatially collocated with the flights, another subset that has been only spatially collocated (over the month of September),

but the MODIS climatology dataset seems to be representative for the entire region. Is that right? I am also curious if the authors have investigated what happened on the date of 12/09, that shows anomalous values of the MODIS data? This is important, since this high value seems to skew the mean AOD of that longitude bin in Fig. 8, and implicitly affect the analysis and conclusions of the paper.

*We have added descriptions a bit sparsely through section 2.5 and section 4.3 to address the main comments here. In section 2.5 we now describe the MOD06ACAERO as following:* "[. . . ] *MODIS instruments with a constant aerosol-cloud vertical geometry and two different aerosol intrinsic property model assumptions. The aerosol models stem from either Haywood et al. (2003) or from the standard MODIS Dark Target land Aerosol product, which is the model used in this work (MOD04; Levy et al., 2009).*" *And at the end of this paragraph* "*Note also that for this work the retrievals are aggregated to a 0.1° Âãequal-angle latitude/longitude grid.*"

*For the MODIS standard retrieval of AOD used as climatology, we have rephrased the last paragraph of section 2.5 in this manner:* "*For another comparison, we use the standard Dark Target aerosol retrieval from MODIS clear sky pixels in the SEA that has been retrieving aerosol properties from reflectances measured since 2001 (Levy et al. 2013). We used 12 years of the high-resolution time series of the MODIS retrieved fine mode AOD sampled during August and September as a proxy for an ACAOD climatology similarly to Zuidema et al. (2016). Using the fine-mode total column AOD to represent the smoke aerosol above cloud in this region is supported by the aerosol's typically small size (Pósfai et al., 2003), and is used to exclude the coarse mode aerosol which mostly consists of boundary layer sea salt and dust along the coast. The presence of biomass burning aerosol results in the fine-mode fraction vastly dominating the optical characteristics of above*

*cloud aerosol in the region (e.g., Yoon et al., 2012, and fine mode frac-
tion by volume in Russell et al., 2014). When there is a significant amount
of biomass burning aerosol in the boundary layer in addition to the aerosol
above cloud, this fine mode assumption is expected to be an overestimate."*

*For the colocation and details of the ACAOD measured by 4STAR and the
retrievals from MODIS, section 3.3 (now 4.3), first paragraph has these de-
tails added: "[. . . ] from the NASA P-3 to those retrieved from MODIS satel-
lite measurements (both standard aerosol Dark Target and above cloud
retrievals)." , and the following sentence: "We focus on the diagonal rou-
tine flight paths (southeast to northwest), where the P-3 sampled the same
locations numerous times over the course of the month-long deployment,
and the MODIS pixels within 15km of the P-3 tracks." . The second para-
graph now reads: "We compile daily 4STAR ACAOD and MOD06ACAERO
values to a mean and median (spanning the August - September 2016 OR-
ACLES deployment period), which we then compare to a proxy of ACAOD
climatology based on the standard MODIS Dark Target fine mode aerosol
retrieval (Fig. 9b $\&$ 9c). The ACAOD proxy is the monthly-averaged MODIS
fine mode AOD for clear-sky pixels that have been aggregated from its orig-
inal high resolution to $1°$ in latitude and longitude following the diagonal rou-
tine flight track of the P-3. The above cloud aerosol is fine-mode dominant
(Sect. 4.2), while the boundary layer aerosol is coarse mode dominant..
[. . . ]"*

*The anomalous data on 12/09 have been investigated, and there is no
clear evidence from satellite measurements that there are problematic val-
ues, only the highest quality assured data is presented. However, we do
propose possible causes for discrepancies in the newly added last para-
graph of section 3.3 (now 4.3), which reads:" Additionally, the filtering of*

*MOD06ACAERO, to only used retrievals over opaque water clouds (with optical thicknesses greater than 4), may lead to systemic biases in ACAOD. Aerosol embedded within clouds have been shown from spaceborne polarimeter measurements to skew ACAOD retrievals (Deaconu et al., 2017). Although based on different retrieval principles, having embedded aerosol within clouds would likely produce a similar reflectance spectral in MODIS measurements than aerosol above clouds, leading to biased high retrievals of ACAOD that includes the optical impact of cloud-embedded aerosols.*"

Another concern is related to the analysis presented in section 3.5. It is not clear to me if this analysis was made along the routine flight track or using the entire dataset of measurements. If the samples are averaged along longitude bins from all the measurements, wouldn't that bias the results due to temporal and spatial variations of the vertical profile of the aerosol layer? The measurements were made on the extent of several days, in which the meteorological state or even diurnal cycle could have impacted the aerosols transport and their altitude. The authors should be more careful in making a statement on the expected aerosol layer transport and distance to cloud, particularly because of the meteorological uncertainty.

*To make it more clear that the analysis of section 3.5 (now 4.5) is pertinent to the entire region we have added these comments in the second paragraph: "Unlike previous studies from spaceborne lidars (Devasthale and Thomas, 2011; Rajapakshe et al., 2017), we found that the gap does not linearly decrease towards the west in a near-monotonic fashion, within the entire region sampled by the NASA P-3 (Fig. 14). Figure 14a shows the meridionally averaged gap extent for all the samples, convolving the temporal and latitudinal variations." To bring caution regarding the importance of the meteorological state of this analysis, we have added this comment:*

*"The smallest gap extent is observed at longitudes westward of 2.0° E, similarly to CALIOP measurements (not shown; Wood et al., In prep.), but may be biased due to the low number of days sampled (only a maximum of 3 days, with 6 different profiles) resulting in a relatively large impact of the meteorological state comparatively to the driving impact of the climatology"*

In my opinion, the paper could benefit from a little restructuration. Firstly, I consider that the study case can be presented as a methodology case, that leads further to the AE computation. The method of computing AE should also be presented in the methodology. I would also gather the AOD and AE sample statistics in one section, and the averaged spatial distribution of AOD and AE in another section. From my standpoint, this structure would make the paper easier to follow and clearer in its scientific message.

I suggest here a new plan, that the authors could consider.

1. Introduction

2. Data and instrumentation

2.1. ORACLES (Fig 1)

2.2. 4STAR

2.3. In-situ instruments

2.4. AERONET

2.5. Satellite retrievals

3. Methodology

3.1. AOD above cloud determination (Fig 13 + Fig 2)

3.2. Spectral AOD above cloud and AE
3.2.1. Hyperspectral AOD study-case (Fig 11 + Fig 12)
3.2.2. AE estimation (Fig 5)

4. Results

4.1. Statistics of sampled AOD and AE (Fig 3 + Fig 6)

4.2. Spatial distribution of AOD and AE (Fig 4 + Fig 7 + figure with number of measurements
/ grid box)

4.3. Airborne AOD in context of climatology and satellite measurements (Fig 8)

4.4. Aerosol vertical profiles and distance to clouds

4.4.1. Spatial variability in AOD profiles (Fig 9)

4.4.2. AE vertical dependence (Fig 10)

4.4.3. AOD distance to cloud (Fig 14)

5. Summary and discussion

*We appreciate the effort to restructure this paper, but we do not interpret the*

*case-study as a methodology. We therefore have restructured the paper, to bring in Fig. 13 (now fig. 2) to the AOD above cloud determination, which is now in a separate methodology section (section 3), alongside the AE calculation, which seems to be inline with your suggested comments. Please see the revised document for the new structure and position of figures.*

Specific remarks:

Page 5, lines 5-10: The absorption coefficient is taken in dry conditions, while the scattering coefficient is measured at different relative humidity conditions. How does that impact the computation of the extinction coefficient and what is the resulted uncertainty?

*This question continues to be an open discussion in the scientific literature and is well outside the scope of this paper. The differences between the extinction coefficient from in situ calculations using dry absorption, and humidified scattering and the extinction from 4STAR, measuring ambient aerosols, may be considered to address this question, but it is outside the scope of this manuscript.*

Line 43: What is the minimum cloud optical thickness for which the MODIS product can retrieve AOD above cloud? Do you know what are the uncertainties of the MODIS ACAOD product for optically thin clouds (e.g. COT < 5)?

*We included at the end of this paragraph these extra details from Meyer et al., 2015: "Meyer et al. (2015) showed MOD06ACAERO retrieved cloud*

*optical thicknesses and effective radius are consistent in range and values with the standard MODIS cloud products, and larger than the standard above cloud AOD product from the spaceborne CALIOP. Consistent with Meyer et al. (2015), we report only the AOD above clouds from MOD06ACAERO with an optical thickness of greater than 4, and AOD uncertainties lower than 100% ."*

Page 6, line 5: "two aerosol model assumptions... " What are those assumptions? Lines 16-20: Maybe move Page 14, lines 8-11 here for better clarification. Do you know how the measurement uncertainty (60 m) for "no gap" situations affect the ACAOD retrieval?

*We included information from Meyer et al., 2015 into this sentence to clarify this description. The new sentence is "Retrievals are run on both Terra (morning) and Aqua (afternoon) MODIS instruments with a constant aerosol-cloud vertical geometry and two different aerosol intrinsic property model assumptions. The aerosol models stem from either Haywood et al. (2003) or from the standard MODIS Dark Target land Aerosol product, which is the model used in this work (MOD04; Levy et al., 2009)." Discussions of gap distances are omitted in the description of this satellite product since it is not addressed in Meyer et al 2015. There is also no evidence of a dependence in the following analysis.*

Page 7, lines 13-15: "Considered together, the ACAOD and full column AOD (denoted by the total extent of the histogram bars in Fig. 3) represent what a satellite remote sensor would retrieve in the region, if it were spatially and temporally co-located to the NASA P-3 aircraft." I understand this sentence, but I don't understand its relevance to the paper.

*This sentence is meant to invite future satellite product comparisons based on this result.*

Page 9, line 3: Why are you fitting the AE up to 1650 nm, since you have showed that for wavelengths larger than 1000 nm, the AOD has higher uncertainties (Fig. 5)?

*Both following the cited literature, O'Neill et al., 2001 and Shinozuka et al., 2011, and because combining many more measurements past 1000 nm actually reduces the computed uncertainty in the polynomial fit, even though the inherent uncertainty in those AOD are larger. Amended the citation from "e.g., O'Neill . . . " to "similar method to O'Neill . . . " to make the link clearer.*

Lines 25 – 27: You mention the negative difference of above cloud AE500 -AE470/865 of -0.26 as representative to biomass burning sources, as defined by Yoon et al., 2012. I do not understand why the full column difference is positive? Aren't you subtracting AE470/865 (=1.25) from AE500 (=1.08) which would result in -0.17? I am confused of your conclusion for this analysis. Could you please clarify it in the text?

*There was a missing negative sign ahead of the -0.17 value. This has been rectified. The values of AE difference are to be considered together with the values of the AE, which is showcased in Yoon et al., 2012. We have amended the text to better describe this. It now reads: "The difference in average AE evaluated at different wavelengths, (AE500 - AE470/865) is -0.26 for the ACAOD, which is very similar to the combination of AE470/865 and AE difference (centered at an AE difference of -0.2, and AE470/865 of 1.85) sampled by the Mongu AERONET station within the biomass burning*

*source region of southern Africa (Yoon et al., 2012). The full column average AE difference of -0.17 with an AE470/865 of 1.25 is typical of coarse-mode dominant, with Mie theory predicting 30% – 40% of fine mode fraction for this combination of AE difference and AE values (Yoon et al., 2012). This large coarse-mode fraction is corroborated by the in situ measurements of large marine aerosol particles during the boundary layer flight segments during ORACLES, or reports of local dust in the boundary layer sampled at the AERONET Mongu station.*"

Page 10, line 33: Why are you using August in the MODIS AOD climatology? As shown is Adebiyi et al., (2015) there is a shift in meteorological condition between August and September, and this would result in a change in aerosol transport – hence climatology.

*Measurements span the month of August and September, amended the paragraph to remind the reader.*

Page 12, Section 3.4.1: - Figure 9 c): how come there are profiles that begin almost at 0 m? Shouldn't these be measurements above the cloud? Are these all column AODs or ACAODs?

*Clarification within the introductory sentence of Figure 9 (now fig. 10) has been made to indicate:* "*with the vast majority representing the ACAOD profiles, and some representing full column profiles*"

- "[. . . ] the variability of the AOD profile in these different regions, we observed at 2000 m AOD ranges between 0.17 to 0.6 for 25 profiles along the routine diagonal, and 0.3 to

0.58 for coastal profiles." Are you discussing only figures 9c) and 9d)? If you consider all the plots, then at 2000 m I observe larger values of AOD closer to the coast.

*This description does refer to Figure 9c and 9d (now 10c, 10d), the text now refers to "the southern profiles" .*

Page 12, Section 3.4.2.: I do not understand how there are more data at 3000-4000 m for "all data" situations in Figure 10 compared to "above cloud" situations. Wouldn't the measurements at that altitude include only above-cloud measurements? Why is the AE above cloud profile stopping at 4000 m, when the ACAOD goes up to 6000 m (Fig. 9)? All the AOD above 4000 m is < 0.1, and thus you have filtered these data?

*The ACAOD is only a subset of alldata representing the entire column of aerosol above cloud, which is flagged using the method described in section 2.6 (now 3.1). All data includes partial columns, of which a smaller portion is included in the ACAOD subset. Slight clarification is made in the Figure 10's (now fig. 11) caption.*

Figure 1: Add the latitude and longitude coordinates

*Figure updated.*

Figure 14 is confusing. You talk about profiles in the text, but show days on the plot. You should mention one or the other in the text or on the plot.

*Switched to sampled days, but included the related number of profiles in the text, kept number of sampled days in the Figure 14 caption.*

Technical corrections:

General observation: I observed missing commas in many places, which makes the text difficult to follow in some cases.

*Commas added.*

Page 2, line 41: Wen et al., (2017) reference is missing. I also suggest Cornet et al., (2018) that looked at POLDER 3-D cloud radiative effects.

*Wen 2007 reference added. Cornet et al., 2018 citation added.*

Page 3, Line 1: You could also mention Hu et al., (2007) for the CALIOP Depolarization Ratio Method, since you mention Chand et al., (2009) for CALIOP Color-Ratio Method.

*Added citation.*

Line 1: Matus et al. 2015 reference is missing

*Reference was adjusted. (It was wrongly formatted)*

Lines 11-15: Maybe split this sentence, as it is hard to follow: "Past work has shown that the elevated aerosol layers in this region are frequently separated from the underlying cloud top, e.g., Devasthale and Thomas (2011) found that 90-95

*Sentence split.*

Page 5, line 1-3: Rephrase: "with paired single-wavelength nephelometers (Radiance Research M903 measuring at 540 nm, with air in one humidified to 80

*Rephrased to: "(two - Radiance Research M903 measuring at 540 nm; one with air humidified to 80% relative humidity, and the other did not control the RH)"*

Line 31: You have already mentioned Gobabeb and Henties Bay in line 26. Maybe rephrase a bit so you don't repeat the same information.

*Rephrased*

Page 6, Line 3: "...uses reflectance ..."

*Changed.*

Lines 40-42: Change "[. . . ] including the biomass burning layer of aerosol above clouds as well as any lower-level aerosol near the sea surface." to "including the elevated biomass burning layer as well as any lower-level aerosol near the sea surface."

*Changed.*

Page 9, Line 11: Kaufman 1993, missing reference

*Added.*

Line 42: "[. . . ] in the southern part of the sampling" should be correct

*Corrected.*

Page 11: lines 11-13: "The peak August and September mean ACAOD from MOD06ACAERO at the most western edge of the region, near 0 E, is shifted to the east in the subsampled MOD06ACAERO." – unclear of which product you are referring to

*Modified to* "*The peak mean ACAOD for all August and September MOD06ACAERO at the most western edge of the region, near 0° E, is shifted to the east in the mean MOD06ACAERO subsampled for routine flights.*" .

Lines 15-18: this sentence is too long "The largest [. . . ] monthly statistics"

*Sentence split.*

Page 12, line 20-22: "[. . . ] large variability is noticeable, especially when contrasting the far-from-coast versus profiles along the routine diagonal" . High variability of? Also, I thought far-from-coast is along the routine diagonal, thus this sentence is not clear.

*Changed the typo from 'far-from-coast' to 'near-coast'.*

Page 14, line 5: there is no Section 3.2.3

*Corrected to say section 3.1.*

Line 12: Sakaeda et al., 2011: This study does not analyses the aerosol-cloud distance (longitudinal gap) from spaceborne lidar. Could you replace your reference with the right one?

*Corrected to say Devasthale and Thomas, 2011*

Page 15, line 15: add reference for the MODIS AOD 12 years climatology

*Amended section 2.5, to illustrate the origins of the climatology. Now the last paragraph says: " For another comparison, we use the standard Dark Target aerosol retrieval from MODIS clear sky pixels in the SEA that has been retrieving aerosol properties from reflectances measured since 2001 (Levy et al. 2013). We used 12 years of the high-resolution time series of the MODIS retrieved fine mode AOD sampled during August and*

*September as a proxy for an ACAOD climatology similarly to Zuidema et al. (2016)." We prefer to keep the references in this part of the paper as to not overburden the discussion.*

Line 23: "For the full column AOD, the AE470/865 is much lower than its above cloud counterpart" . It is not much lower, since the value of the mean difference between ACAOD and AOD equals 0.4, which is within the uncertainty range (standard deviation of ACAOD is up to 0.4 – Figure 7).

*Reframed the sentence to focus on the area averages, as indicated in Table 1. Now reads: "Looking at the ensemble of the region, Table 1 shows for the full column AOD, the AE470/865 is lower than the AE from ACAOD, this is more evident when considering the spatially binned AE from full column AOD vs. ACAOD, which are well outside one standard deviation from their respective means. This notion is also supported by the vertical profile of AE (Fig. 11) which indicates the presence of large aerosol particles, potentially marine aerosol embedded within the lower boundary layer, only when considering the full column AOD."*

Line 32: You could add a discussion of the following sentence: "The ACAOD from 4STAR also has a peak closer to shore than the MODIS AOD climatology mean and median (both fine and coarse mode), with differences near coast between 4STAR ACAOD measurements and MOD06ACAERO retrievals."

*Added a few sentences of discussions: "Differences between 4STAR ACAOD and the MOD06ACAERO subsampled for the same day are possibly linked with daily airmass movement and underlying cloud diurnal cycle,*

*especially when there is a mismatch between MODIS overpass times and aircraft sampling times. The subsampled MOD06ACAERO is more similar to the August mean average than the September average, which can partially explain the sampling representativeness, and therefore some differences, between 4STAR ACAOD and September climatology built from MODIS measurements.*"

Page 16, lines 10-12: "From these airborne measurements, we have seen that the ACAOD is lower than expected from current MODIS satellite retrievals during the measurement period (by 0.05-0.08) and from a 12-year climatology (by 0.04)." Unclear sentence. Do you mean "... compared to current MODIS satellite retrievals... " ?

*Sentence now reads:*" [. . . ] *we have seen that the ACAOD is lower than expected from subsampled MODIS satellite retrievals (MOD06ACAERO) during* [. . . ]"

**Supplement:**

[revised manuscript text omitted]

---

## Author Comment (AC2) · 21 Aug 2019

Thank you for your interest in this manuscript.

The relative humidity threshold for the ambient outside value for identifying clouds is set to when it is saturated (near 100%) with some acceptable deviation from 100%, depending on situation, and instrumental error. This saturation liked to couds tends to be very obvious during ascent and descent of the aircraft, therefore the exact relative humidity threshold is not important for these cases.

The same amount of measurements is found at longer wavelengths than the shorter ones. The spectrometers at longer wavelengths have simply a lower sensitivity, there- fore lower signal to noise. But large particles, with a similar AOD at low wavelength,

would disproportionally increase AOd at longer wavelength, giving a larger standard deviation.

Restructured this whole section, which now reads "There is a prevalence of near zero gap extent, while the largest gaps extents are observed not close to coast, as expected, but off-shore near 7° E.". There is no gap distance reported by MOD06ACAERO. This constant geometry of cloud-aerosol distance is now included in the manuscript as possible source of errors, albeit there is no correlation between the divergence of MOD06ACAERO and the aerosol-cloud gap.

––––––––––––––––––––––––––––

---

## Author Response (AR2)

The authors have made substantial efforts to improve their paper and to take into account the reviewers' comments. There is just the formulation of the new paragraph on page 2 (starting from line 22) that I do not totally agree on. The authors do a list of the uncertainties associated with the retrieval of the ACAOD from passive satellite instrument which I found to be unclear and partly incorrect.

(1) "the assumption that the spectral representation of aerosol absorption is assumed to be constant (e.g., Chand et al., 2009, Meyer et al., 2015)"; "aerosol properties don't vary in a large spatial swath (e.g., Torres et al., 2012); and/or that the retrieved aerosol properties over highly reflective and opaque clouds are representative of all aerosol (e.g., Hu et al., 2007, Peers et al., 2015)"

All the retrievals based on the color-ratio effect (passive reflectance only measurements such as MODIS, OMI, VIIRS, SeaWiFS, SEVIRI) need to assume the aerosol microphysical properties (size distribution, spectral refractive index) as well as their vertical distribution, which means that the aerosol model is supposed to be representative of the whole region and for the whole biomass burning season. The single scattering albedo is the assumption which produces the largest uncertainty in the ACAOD retrieval.

> *That section now reads:*
> *"Underlying assumptions of aerosol optical and microphysical properties and of cloud properties in retrievals of ACAOD from satellites can lead to large uncertainties or biases. Examples of assumptions include: a constant spectral aerosol absorption, which has the largest influence on retrieval uncertainty (e.g., Chand et al., 2009, Meyer et al., 2015); aerosol properties don't vary over a large spatial region, or are representative of all aerosol over large regions, and have a constant vertical dependence (e.g., Torres et al., 2012); retrieved aerosol properties over highly reflective and opaque clouds are representative of all aerosol (e.g., Hu et al., 2007, Peers et al., 2015); and/or the impact of aerosol absorption on polarized reflectances can be neglected (e.g., Waquet et al., 2013b, Peers et al., 2015). "*

(2) "aerosols only weakly impact polarized reflectances (e.g., Waquet et al., 2013b, Peers et al., 2015)"
Aerosols do impact the polarized reflectances. However, the algorithm based on polarized reflectances do consider that the impact of the aerosol absorption on the polarized signal can be neglected.

> *We have clarified the impact to the reflectances in the revised section above. Now reading: "[...] the impact of aerosol absorption on polarized reflectances can be neglected (e.g., Waquet et al., 2013b, Peers et al., 2015). "*

I would recommend the publication of this paper if these remarks are addressed.

Anonymous referee #2

The paper written by Samuel LeBlanc et al. is focused on the Aerosol Above Cloud (AAC) properties measured by the 4STAR instrument during the ORACLES 2016 airborne campaign. The magnitude and spatial variability, as well as the vertical distribution of above-cloud aerosol optical depth (ACAOD) and Ångström Exponent (AE) are discussed. A comparison of 4STAR ACAOD with satellite retrievals from MODIS of clear-sky fine mode AOD and ACAOD is also made. Their results show that during the ORACLES 2016 campaign the aerosol properties sampled above the clouds are consistent with previous studies, which show high elevated thick layers of fine particles, typical to biomass burning aerosols. They also found that the largest ACAOD is found in the northern part of the sampling region, and that high AOD variability coincides with high AE variability. Both the satellite climatology over August and September and MODIS ACAOD retrievals collocated along the flight tracks consistently overestimate the 4STAR ACAOD. The final results show that the gap between the aerosol layers and the cloud tops is larger further away from the coast compared to near-coast samples.

The paper is now well documented and well written. The authors have made a wholesome effort in giving detailed responses to the reviewer's questions and comments, modifying the text where needed, which increased the scientific value of the paper and concluded an in-depths study of aerosols above clouds over the Atlantic Ocean.

In my opinion, the paper is ready to be published in ACP.

Specific remarks:
Page 4, line 8: modify with Deaconu et al., 2019
> *Ammended*

Page 9, lines 12-14: Are the maximum values for the absolute range for ACAOD and full-column AOD correctly provided, since you mention: "
[revised manuscript text omitted]